# Private benefit of β-lactamase dictates selection dynamics of combination antibiotic treatment

Helena R. Ma [1,2], Helen Z. Xu [3,4], Kyeri Kim [1,2], Deverick J. Anderson[5,6] & Lingchong You [1,2,7] ✉

β-lactam antibiotics have been prescribed for most bacterial infections since their discovery. However, resistance to β-lactams, mediated by β-lactamase (Bla) enzymes such as extended spectrum β-lactamases (ESBLs), has become widespread. Bla inhibitors can restore the efficacy of β-lactams against resistant bacteria, an approach which preserves existing antibiotics despite declining industry investment. However, the effects of combination treatment on selection for β-lactam resistance are not well understood. Bla production confers both private benefits for resistant cells and public benefits which faster-growing sensitive cells can also exploit. These benefits may be differentially impacted by Bla inhibitors, leading to non-intuitive selection dynamics. In this study, we demonstrate strain-to-strain variation in effective combination doses, with complex growth dynamics in mixed populations. Using modeling, we derive a criterion for the selection outcome of combination treatment, dependent on the burden and effective private benefit of Bla production. We then use engineered strains and natural isolates to show that strong private benefits of Bla are associated with increased selection for resistance. Finally, we demonstrate that this parameter can be coarsely estimated using high-throughput phenotyping of clonal populations. Our analysis shows that quantifying the phenotypic responses of bacteria to combination treatment can facilitate resistance-minimizing optimization of treatment.

β-lactams are the most commonly prescribed antibiotic in the United States today[1] in both inpatient[2] and outpatient[3] settings. This broad class, which includes the penicillins, cephalosporins, carbapenems, and monobactams, includes both first-line antibiotics for many outpatient cases[4,5] as well as agents reserved for cases in which other regimens have failed[6]. As with many antibiotics, however, resistance to β-lactams has become prevalent as an evolutionary consequence of their use. Of note is the steadily increasing incidence[7–9] of resistant

infections by the Gram-negative *Enterobacteriaceae*. Resistance in these infections is associated with higher mortality, delay in effective therapy, and higher healthcare costs[10]. With the novel antibiotic pipeline thinning due to several economic and scientific factors[11,12], maximizing the efficacy of existing antibiotics is key to minimizing the impacts of resistance.

One strategy to potentiate existing antibiotics against resistant bacteria is to pair them with adjuvants that can inhibit the mechanism

[1]Department of Biomedical Engineering, Duke University, Durham, NC, USA. [2]Center for Quantitative Biodesign, Duke University, Durham, NC, USA. [3]Department of Biology, Duke University, Durham, NC, USA. [4]Department of Computer Science, Duke University, Durham, NC, USA. [5]Division of Infectious Diseases, Department of Medicine, Duke University School of Medicine, Durham, NC, USA. [6]Duke Center for Antimicrobial Stewardship and Infection Prevention, Duke University School of Medicine, Durham, NC, USA. [7]Department of Molecular Genetics and Microbiology, Duke University School of Medicine, Durham, NC, USA. ✉e-mail: you@duke.edu

of resistance[13–15]. Resistance to β-lactams in Gram-negative bacteria is primarily mediated by β-lactamase (Bla) enzymes, which hydrolyze the β-lactam ring and inactivate the antibiotic[16]. Bla inhibitors can bind to the active site of the enzyme, preventing hydrolytic activity and restoring antibiotic sensitivity. They have been used in combination with β-lactams since the discovery of clavulanic acid in the 1970s[1,16]. Inhibitors with activity against more diverse β-lactam classes, such as the recently approved durlobactam, have since been developed[13,17].

While Bla inhibitors permit clinical treatment of resistant populations, the use of β-lactam/Bla inhibitor combinations, like many antibiotics, is empirical. The diagnosis of common infectious diseases is often done by symptom rather than by infectious agent; when specific bacteria are diagnosed, it may be done only at the species level; and when antibiotic resistance is screened, it is typically done with only a final timepoint measurement or single-gene marker for resistance, obscuring dynamics that affect the course of treatment[18]. In prescription, the most relevant differences between agents that perform roughly the same function, such as the different Bla inhibitors clavulanic acid, sulbactam, and tazobactam, seem to be their pharmacokinetic similarity to the β-lactams they are formulated with[19]. This coformulation, a consequence of the regulatory framework for combination drugs, also limits the flexibility with which the two components (antibiotic and Bla inhibitor) can be dosed. These realities all lead to a one-size-fits-all approach to antibiotic administration.

However, strains are not all equal. For instance, in previous work, we demonstrated how single-cell- and cooperative population-level traits which allow bacteria to survive antibiotic treatment have different impacts on post-treatment population dynamics, and that clinical isolates vary in these traits[18]. By understanding how these differences affect treatment outcomes, we can better optimize treatment for situations and objectives that are not currently well accounted for.

A particularly important objective is to minimize selection for resistance. However, the selection dynamics of β-lactam treatment when both resistant (Bla-producing) and sensitive strains are present can be complex. In addition to the private benefit of reduced antibiotic susceptibility for the producing cell, sensitive cells are partially protected through antibiotic degradation by resistant cells, making Bla production a public good[20,21]. Cells not participating in this cooperation can exploit their Bla-producing neighbors by reaping the benefits without contributing, allowing for both subpopulations to survive treatment. Bla inhibitor use can modulate the direction of selection by diminishing this protection. In fact, the addition of a Bla inhibitor alongside a β-lactam was found to increase the fraction of resistant cells in a mixed population[20], suggesting that the use of Bla inhibitors could promote the spread of β-lactam resistance. Yet, a separate study showed that strains selected for under a high treatment ratio of inhibitor to antibiotic were subsequently more sensitive to the antibiotic than strains selected for under a low treatment ratio of inhibitor to antibiotic[22], suggesting that Bla inhibitors could instead reduce selection for resistance. This apparent paradox highlights a need to understand the factors that govern the selection dynamics of bacteria responding to combination β-lactam/Bla inhibitor treatment.

In this work, we use high-throughput generation of growth curves to demonstrate how β-lactam/Bla inhibitor killing dynamics vary across strains. We then use quantitative modeling to dissect the selection dynamics of a mixed population of sensitive and resistant bacteria responding to combination treatment. We demonstrate how both dose and strain-specific properties affect selection dynamics and derive a simplified criterion to predict them. Using engineered strains and clinical isolates, we then demonstrate how the outcome of combination treatment with β-lactam/Bla inhibitor combinations depends generally on the degree to which resistance is private rather than cooperative, and show how phenotypic quantification of clonal strains can predict the strength of the private benefit and the consequent selection dynamics.

## Results

### Strain- and drug-based variation in high-throughput dose response landscapes

To survey the limitations of a one-size-fits-all approach to combination dosing, we measured the response of *Escherichia coli* strains of different genetic backgrounds or expressing different Bla enzyme variants to combinations of amoxicillin and different Bla inhibitors (Fig. 1A–C). We first contrasted dose-response landscapes for a resistant laboratory strain and two clinical isolates responding to cefotaxime and clavulanic acid (Fig. 1A), as well as another resistant laboratory strain responding to amoxicillin and the three different Bla inhibitors clavulanic acid, sulbactam, and tazobactam (Fig. 1B). We found that both the dose required to kill a population inoculated at the same initial concentration, and the shape of the boundary, indicating the degree of synergy or antagonism, could differ between strains or Bla inhibitors.

To further explore whether the extent of private benefit from Bla production could affect cell survival, we used laboratory strains expressing either a periplasmic (wild-type) Bla enzyme, which confers single-cell-level protection against β-lactam induced lysis (resistance has a private benefit), or a cytoplasmic Bla enzyme, called BlaM, which is engineered to lack the signal peptide for periplasmic localization. BlaM is released to the population upon lysis but confers reduced protection to the expressing cell. Thus, BlaM resistance is more of a public good[23]. We found that strains with more private benefit (Bla) and faster growth rates (low copy) generally survived higher combination doses than strains with less private benefit (BlaM) or slower growth rates (high copy). We note that for several resistant strains, however, the effective doses were similar after 24 h (Fig. 1C).

The degree of private benefit accrued to producing cells, in excess of the public benefit experienced by the entire population, is known to affect population dynamics. To begin to assess this outcome, we generated dose response landscapes at the same concentrations for a mixture of sensitive (plasmid-free) and resistant (Bla-expressing) cells with initially equal population densities (Fig. 1D)[24]. Mixtures containing strains with higher private benefit exhibited complex, non-monotonic response landscapes: an intermediate region of growth emerged between complete suppression and full growth (Fig. 1D). These patterns were observed for more than one Bla inhibitor (Supplementary Fig. S1). We wondered whether this reflects differential resistance selection in these two-population communities, and if so, whether it was possible to predict the selection dynamics when sensitive strains are cocultured with different resistant strains.

### Modeling selection dynamics in response to combination treatments

To understand the factors governing the selection response to β-lactam/Bla inhibitor combinations, we extend our previous work[18] to develop a kinetic model consisting of ordinary differential equations (ODEs) which describe the dynamics of two populations, one sensitive and one resistant (Bla-producing), interacting with one another in a mixture in response to combination treatment (Fig. 2A). We model multiple processes by which Bla provides private and public benefits. Bla is a public good, benefiting both resistant and sensitive populations in the mixture, in two ways: antibiotic hydrolysis by resistant cells detoxifies the environment for all cells, and the enzyme continues to function after release through lysis or in outer membrane vesicles[25,26]. However, as Bla is primarily localized in the periplasm of the producing cell, it also functions as a private good for resistant cells by reducing antibiotic-mediated lysis[27] (Fig. 2B).

In the absence of β-lactams, both populations grow until reaching an overall carrying capacity dictated by nutrient availability. β-lactams induce concentration-dependent lysis in both populations. Compared with sensitive cells, we assume that resistant cells experience a reduced growth rate, due to the burden of Bla production, and a

reduced lysis rate. In our model, we represent the burden of Bla production as the growth-rate-modulating factor $\alpha$, where $0 < \alpha < 1$. Likewise, we represent the private benefit through reduced lysis as the lysis-rate-modulating factor $\beta$, where $0 < \beta < 1$. Finally, Bla inhibitor reduces both antibiotic degradation by extracellular Bla and resistant cells, as well as the lysis protection afforded to resistant cells. The latter two effects depend on the ability of the inhibitor to penetrate the outer membrane, a parameter that may vary between strains and inhibitors[28,29]. We represent this parameter as $c$, a modulator of

antibiotic degradation by resistant cells and of lysis reduction. If $c = 0$, degradation and lysis reduction for the resistant strain are at that strain's maximum, while if $c = 1$, there is no degradation and no lysis reduction. Finally, if there is sufficient cell survival, over time, the degradation of the antibiotic allows for the recovery of both populations (Fig. 2C).

We simulated the response of this mixed population, where the densities of the sensitive and resistant subpopulations are initially equal, to different initial antibiotic and inhibitor concentrations. With

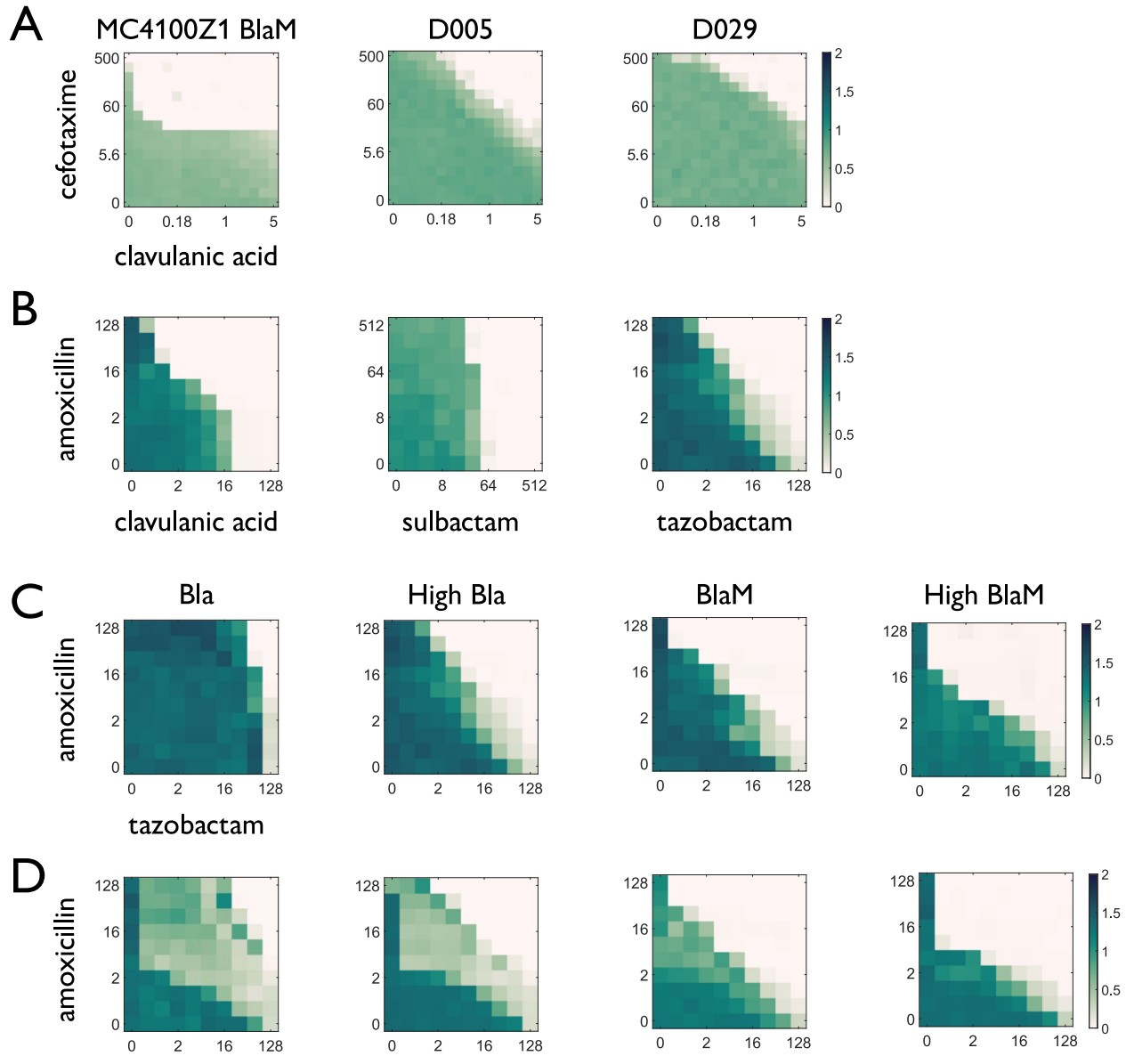

**Fig. 1 | High-throughput measurement of β-lactam/Bla inhibitor dose response matrices.** Color reports average OD600 at 24 h ($n = 3$) unless otherwise specified. **A** Positions (reflecting effective concentrations) and shapes (reflecting synergy or antagonism) of the boundary between cell survival and cell suppression differ between *E. coli* strains of different backgrounds. From left to right, a laboratory strain expressing a version of Bla (MC4100Z1 BlaM) and two ESBL-producing clinical isolates (D005 and D029) were treated with increasing concentrations of cefotaxime (*y*-axis) and clavulanic acid (*x*-axis), both in µg/mL. OD600 at 40 h reported ($n = 5$). **B** Shapes and positions of the boundary between cell survival and cell suppression differ between different beta-lactamase inhibitors. A laboratory strain (DA28102) expressing Bla on a high-copy plasmid was treated with

amoxicillin (*y*-axis) and, from left to right, clavulanic acid, sulbactam, or tazobactam (*x*-axis), all in µg/mL. **C** Strains expressing the wild-type version of Bla enzyme (Bla) or expressing it from a lower-copy plasmid are able to survive in higher concentrations than strains with less private benefit (BlaM) or expressing it from a higher-copy plasmid. From left to right, laboratory strains (DA28102) expressing Bla or BlaM on a low- or high-copy plasmid were treated with amoxicillin (*y*-axis) and tazobactam (*x*-axis), both in µg/mL. **D** Intermediate cell densities are observed for the strains (same order as in **C**) when a mixed population of initially equal sized resistant (plasmid-carrying DA28102) and sensitive (plasmid-free DA28102) subpopulations are treated with amoxicillin (*y*-axis) and tazobactam (*x*-axis), both in µg/mL.

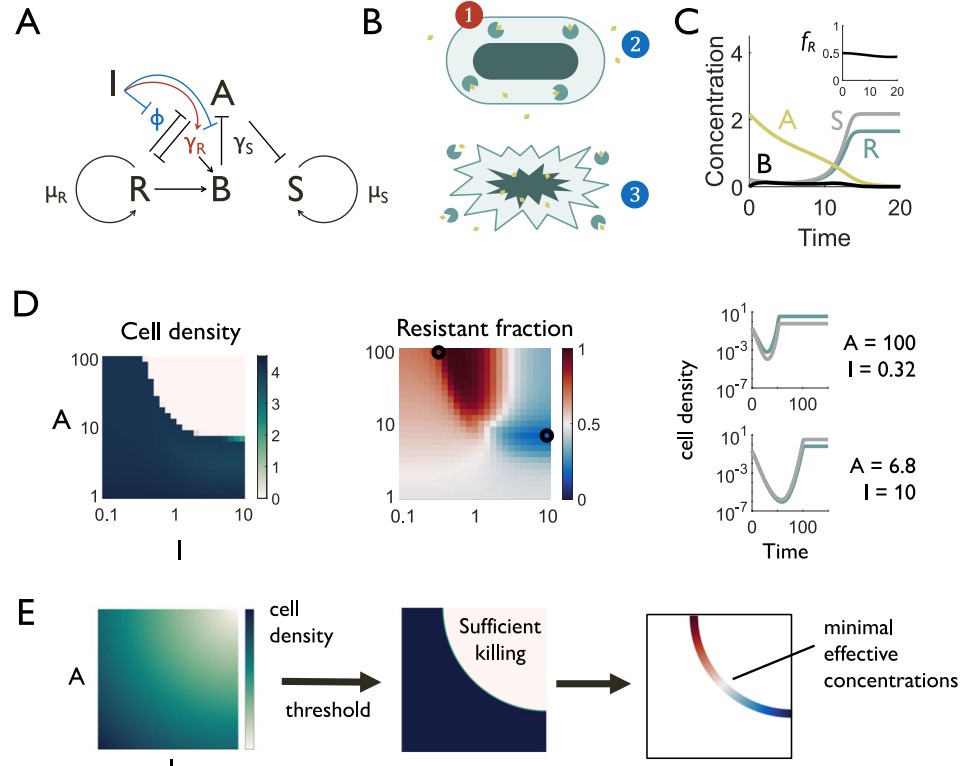

**Fig. 2 | Dynamics of mixed good beta-lactamase production in a heterogeneous population. A** A simple dimensionless mathematical model captures the interactions of sensitive ($S$) and resistant ($R$) populations in the presence of antibiotic ($A$) and Bla inhibitor ($I$), in which resistant cells have both a slower growth rate ($\mu_R < \mu_S$) and a slower antibiotic-mediated lysis rate ($\gamma_R < \gamma_S$). Antibiotic degradation occurs via intact resistant bacteria ($\varphi$) and via free Bla ($B$) released from lysis of resistant cells. Inhibition affects both the private (red) and public (blue) components of Bla-mediated resistance. **B** Bla (teal) is localized to the cellular periplasm and offers private protection (red) to the producing cell (1). However, it also offers public protection (blue) to surrounding cells, which may be sensitive, through two mechanisms: overall reduction in antibiotic (gold) concentration (2) and Bla release upon cell death (3). **C** Simulated time courses of population densities (S, R), antibiotic (**A**), and extracellular Bla (**B**) after a dose of combined antibiotic and inhibitor treatment ($a_0 = 2.15$, $i = 10$). Antibiotic treatment results in the initial death of both resistant and sensitive populations. Over time, resistant cells and free Bla released by resistant cell lysis degrade antibiotic, eventually reducing lysis rates and allowing for the recovery of both populations to a steady state determined by total nutrient availability. Resistant cells may have an advantage while antibiotic levels remain high, but in the absence of antibiotic have a growth deficit relative to sensitive cells. Inset shows the corresponding time course of the fraction of resistant cells $f_R$ in the population. **D** Different doses of antibiotic and inhibitor correspond to different final cell densities (dimensionless, yellow to green color) and resistant fractions (blue to red color). Antibiotic and inhibitor have a synergistic effect, and higher concentrations of either lead to lower cell densities. Example time courses for doses that produce either majority-resistant or majority-sensitive populations are highlighted. **E** Clinical objectives for dose optimization include population suppression, which can be defined by an acceptable threshold, and using minimal drug concentrations within the acceptable range, to minimize cost, resistance, and side effects. Selection against resistance provides a third objective for dose optimization.

all other parameters held constant, depending on the dose, combination treatment could result in both majority-resistant and majority-sensitive populations. To compare the effects of different combination doses, we examined both the cell density and resistant fraction for each dose at a single final timepoint. As in our experimental results, simulation often predicts a synergy between antibiotic and inhibitor: increasing the amount of Bla inhibitor reduces the effective MIC of the antibiotic (Fig. 2D). Additionally, even if two dose combinations have a similar effect on overall cell density, they may result in different final compositions (Fig. 2D, Supplementary Fig. S2).

Major objectives when optimizing a combination dose include suppressing the pathogen and conserving drug use, where the latter reduces cost, side effects and antibiotic exposure in the environment. Given freedom to adjust the relative concentrations of β-lactam and Bla inhibitor, these objectives can be met at a variety of doses. However, our results suggest that the objective of minimizing the resistant subpopulation, a key goal of antibiotic stewardship, further constrains the optimal dose (Fig. 2E). Although the goal is to select doses that sufficiently eradicate bacterial populations, complete eradication is not guaranteed. Bacterial regrowth can occur after treatment due to incomplete compliance with treatment

prescriptions[30,31] or recurring infections such as those associated with cystic fibrosis[32], and increased resistance in the regrown populations is a clinical challenge. More broadly, antibiotics are frequently released into the environment[33], influencing selection even at subinhibitory levels. Thus, it is important to understand whether and when dose combinations can be designed to select against resistant subpopulations, where repeated treatment with such a dose could then extinguish the resistant population.

## The private benefit of Bla plays a critical role in the outcome of selection dynamics

We varied model parameters one at a time to probe their effects on community survival and composition. Most of the parameters did not strongly affect the community dynamics in terms of selection for the resistant or sensitive subpopulation (Supplementary Fig. S3), except for three. Specifically, the simulation predicted that increasing the burden of resistance ($1 - \alpha$) and reducing the private benefit of Bla ($1 - \beta_{min}$) reduced the conditions under which the community could survive (Fig. 3A). This was also true of increasing intracellular Bla inhibition ($c$), which reduces the private benefit of Bla. The simulation also revealed that these modest effects on community survival were

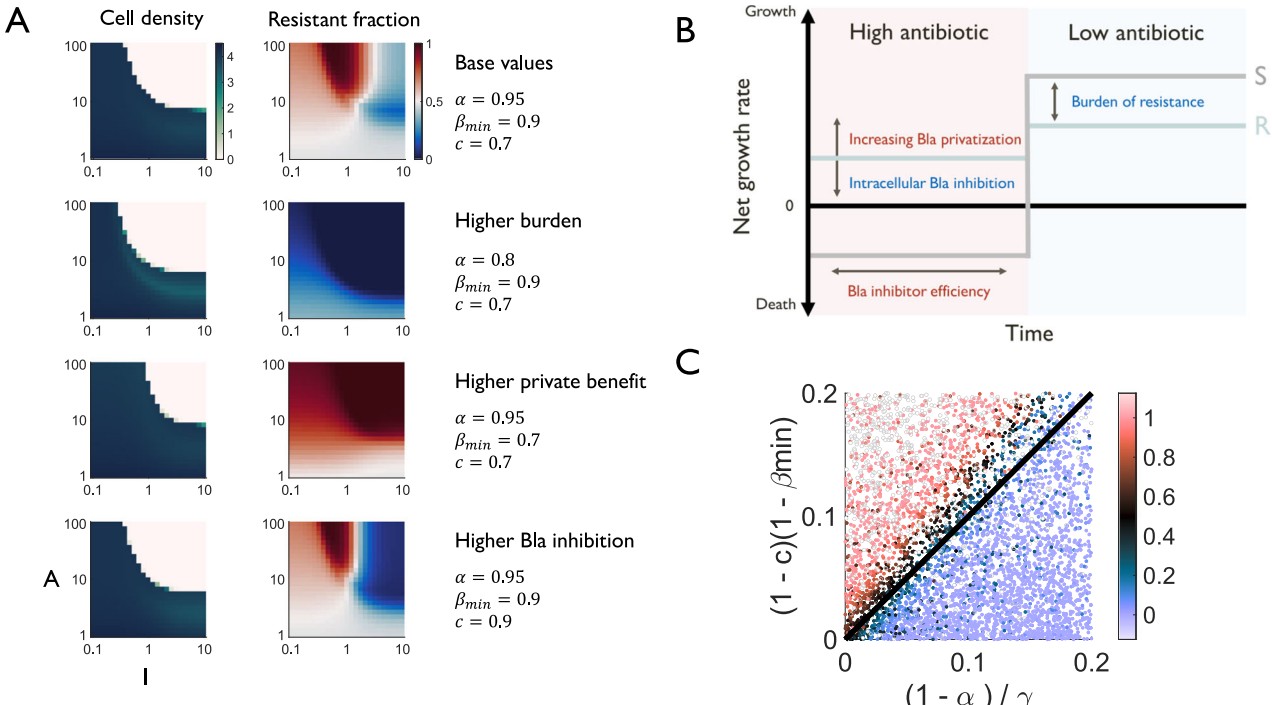

**Fig. 3 | Major strain-specific factors govern evolutionary response for simulated strains. A** All else being equal, higher burdens (lower $\alpha$), lower private benefits (higher $\beta_{min}$), and higher inhibition of intracellular Bla (higher $c$) are associated with a modest decrease in survivable dose concentrations (dimensionless density reported in yellow to green color) and lower final resistant fractions (resistant fraction reported in blue to red color) at all concentrations. Quantities are dimensionless. **B** Drivers of the selection dynamics. When antibiotic concentration is high, resistant cells have a growth advantage due to private benefit, which can be suppressed by Bla inhibition. When antibiotic concentration is low, sensitive cells have a growth advantage due to the burden of Bla expression. The length of the two

periods is governed by the doses of antibiotic and Bla inhibitor as well as antibiotic degradation. **C** The simplified criterion $(1 - c)(1 - \beta_{min}) > (1 - \alpha)/\gamma$, where $\gamma$ represents the maximum lysis coefficient, predicts the outcome of selection dynamics for 10,000 simulated strains with randomized parameters. For each parameter set, we queried each inhibitor concentration and identified the lowest antibiotic concentration for which the cell density was suppressed below 1. We took the lowest resistant fraction in this set as the minimum achievable resistant fraction for doses high enough to suppress the population. The color of each circle reports this value for the corresponding parameter combination. Empty circles indicate that no treatment concentrations achieved sufficient suppression.

often associated with large differences in the selection dynamics. As is well-recognized, increasing the burden of Bla production selected against the resistant subpopulation[34–36].

Our simulation also revealed a critical role of the private benefit of Bla production as well as the impact of modulating this private benefit. Depending on the magnitude of both the private benefit and burden of resistance, resistant cells may have an advantage while antibiotic levels remain high and lysis is dominant. After sufficient antibiotic degradation, however, the driving force of selection dynamics is the growth burden of resistant cells, and the sensitive subpopulation is favored. The overall dynamics are thus determined by both the degree of resistant cells' advantage and the relative length of these two periods, which is governed by treatment concentrations and enzyme inhibition efficacy (Fig. 3B). Increasing the private benefit (decreasing $\beta_{min}$ or decreasing $c$) will enable the resistant population to have a greater advantage over the sensitive population during antibiotic treatment. Conversely, decreasing private benefit (increasing $\beta_{min}$ or $c$) would make resistant cells more similar to sensitive cells during treatment, reducing resistant advantage and favoring the sensitive population. Indeed, increasing $\beta_{min}$ or $c$ led to reductions in the resistant fraction at almost all drug concentrations (Fig. 3A).

These parameters reflect phenotypic traits that can vary from strain to strain. The burden of resistance can arise from the cost of maintaining plasmids, expressing resistance genes, or interactions with other parts of the cellular network[35]. The possibility of exploiting fitness costs to manage drug-resistant populations is well recognized[37–39]. In contrast, the selection impacts of private benefit and intracellular inhibition are less appreciated. The private benefit

of Bla can differ due to enzyme kinetics[40] or changes in localization, such as a propensity for secretion[25]; the degree of intracellular inhibition depends on the membrane permeability[28,29] and enzyme kinetics[16] of the inhibitor. For intracellular inhibition, differences in both genetic properties, which influence membrane properties, and the drug chosen itself can influence this parameter[28,29], and additional adjuvants could even be considered to further enhance inhibitor penetration[15,41,42].

Given these predictions, we aimed to develop a generic criterion for this selection response. Whether sensitive or resistant cells are enriched depends on the relative growth rates between the two: a combination treatment enriches the resistant population if and only if the net growth rate of resistant cells $\rho_r$ is higher than the net growth rate of sensitive cells $\rho_s$. This constraint leads to a general criterion that predicts the selection dynamics (Eq S13):

$$1 - \beta > \frac{1 - \alpha}{l/g} \tag{1}$$

which depends on burden $(1 - \alpha)$, effective private benefit $(1 - \beta)$, and to the lysis rate relative to the growth rate $(l/g)$. This criterion has an intuitive interpretation: The left-hand-side is the effective private protection by Bla (with or without the Bla inhibitor). The right-hand-side indicates the burden experienced by the Bla-producing cells normalized with respect to the maximum net lysis rate. Thus, the criterion states that the resistant population will be selected for when the degree of private protection outweighs the effective burden in producing Bla.

The criterion captures the interplay of private and public benefits as the two populations grow and modulate their chemical environment. It is applicable regardless of how the lysis rate depends on the antibiotic concentration, which in turn is determined by Bla-mediated degradation, thus reflecting the public benefit of Bla. Similarly, the criterion is applicable regardless of how the Bla inhibitor affects the private benefit, which depends on both the concentration and effectiveness of the Bla inhibitor.

To facilitate application of the criterion, we derive a more simplified version by considering the limiting case of saturating antibiotic and inhibitor concentrations (e.g., before substantial antibiotic degradation has occurred). Under these conditions, the criterion is determined by several parameters that are uniquely associated with individual populations and are not affected by the temporal dynamics of the mixture (Supplementary Information):

$$(1-c)(1-\beta_{\min}) > \frac{1-\alpha}{\gamma}, \qquad (2)$$

where c describes the effectiveness of the Bla inhibitor, $1 - \beta_{\min}$ is the maximum private benefit of Bla, and $\gamma$ is the maximum lysis coefficient.

To evaluate the effectiveness of the simplified criterion (Eq. 2) in predicting selection dynamics in general, we simulated combination treatment of ten thousand randomized parameter sets. For each parameter set, we identified the minimum possible resistant fraction achievable within our dose range. Despite our simplifying assumptions, we found that the criterion is an excellent predictor of the selection dynamics for our simulated strains (Fig. 3C). The lack of complete alignment between equality in the criterion ($y = x$ line) and dynamics that do not result in selection for either resistant or sensitive populations (black dots) reflects those assumptions. However, the discrepancy in prediction errs on the conservative side: that is, it overestimates the risk that combination treatment may select for resistance for some strains, rather than underestimating the risk. We found that the predictive power of the criterion was insensitive to initial composition, nutrient availability, and the introduction of non-growth-dependent lysis (Supplementary Fig. S4). We also found that increases in Bla-mediated antibiotic degradation, or the public benefit of Bla production, increased the range of concentrations at which the total population survived and at which sensitive cells dominated, but did not affect the ability of the criterion to predict whether it is in fact possible to select against resistance (Supplementary Fig. S5).

Our base model assumes that changes in resistant and sensitive population sizes are due primarily to growth and lysis, rather than horizontal gene transfer, by which resistant cells can transfer resistance genes to sensitive ones[43,44]. If we include plasmid transfer via conjugation and plasmid loss via segregation error, the criterion becomes (Supplementary Information):

$$(1-c)(1-\beta_{\min}) + \frac{\eta(n_{r0}+n_{s0})}{\gamma g_0} > \frac{1-\alpha}{\gamma} + \frac{\sigma(n_{r0}+n_{s0})}{n_{s0}\gamma g_0} \qquad (3)$$

where $\eta$ is the plasmid transfer rate, $\sigma$ is the plasmid loss rate, and the initial cell densities ($n_{r0}, n_{s0}$) and growth rate ($g_0$) appear. Therefore, when conjugation is present, plasmid transfer rates higher than loss rates favor selection for resistant cells. This result highlights the importance of developing engineering solutions to minimize transfer and maximize loss.

**Quantifying selection dynamics in engineered bacteria**

To experimentally test the predicted selection dynamics, we engineered four strains to tune the burden and extent of private benefit, where we express Bla or BlaM in *E. coli* DA28102 cells using vectors with moderate (p15A) or high (pUC19) copy numbers (Methods). The Bla strains experience a greater private benefit than the corresponding

BlaM strains with the same plasmid copy number. This property is reflected in the crash-and-recovery dynamics of both BlaM strains in response to 50 µg/mL carbenicillin (Fig. 4A). With the same enzyme (Bla or BlaM), the high-copy versions experience a greater burden, as reflected by a greater time lag in growth than their moderate-copy counterparts with or without antibiotic treatment. DA28102 cells not carrying these plasmids (labeled as WT) did not survive in the presence of carbenicillin.

To quantify the effects of combination treatment on the composition of experimental communities varying in our key parameters, we introduced a fluorescent marker (sfGFP) to our Bla and BlaM-expressing plasmids. The host DA28102 strain into which they were transformed contains a different chromosomally integrated marker (mTagBFP2). This allowed us to differentiate plasmid-free sensitive cells from plasmid-carrying resistant cells and to quantify the resistant subpopulation as a fraction of the total live cell population using fluorescence microscopy (Fig. 4B).

We measured the selection dynamics by starting from a mixture of equal amounts of sensitive and Bla-expressing cells, in the presence of 2 µg/mL amoxicillin (slightly above the MIC of the sensitive cells) and increasing concentrations of a Bla inhibitor. For both high- and low-copy-number plasmids and for three different Bla inhibitors, the resistant fraction for BlaM-producing strains was consistently lower than that of Bla-producing strains (Fig. 4C). That is, the combination treatment led to stronger selection for the resistant population when Bla was more privatized. When exposed to the Bla inhibitors alone, Bla- and BlaM-producing strains did not exhibit consistent differences in their selection dynamics (Supplementary Fig. S6).

We note that, even for BlaM cells, the resistant fraction increased with Bla inhibitor in most cases. Although the private benefit is less than that of Bla, BlaM may still provide private benefit for producing cells. Even a moderate private benefit by BlaM (though less than that of Bla) could maintain the inequality in the criterion (Eq. 1) unless the burden is too high.

**Predicting selection responses from clonal measurements of clinical isolates**

Our model predictions are generalizable to any situation in which a resistant population exists in coculture with a sensitive, faster-growing population. Infections in the body do not consist of isolated pathogens but rather those pathogens interacting with a complex mixture of other pathogens and/or resident microflora in a polymicrobial setting. Interactions between these different microbes are known to influence their susceptibility to treatments and their effect on the patient[45-49]. If we consider the resident flora collectively to be a sensitive sub-population, we expect these dynamics to be broadly applicable across many infection contexts. Given that the parameters governing selection dynamics include strain-specific factors, identifying the strains that will be sensitive to β-lactam/Bla inhibitor combinations, in a way that minimizes resistance, can help clinicians choose appropriate treatments for different patients.

Measuring these parameters is nontrivial for clinical isolates. For example, the effective burden of resistance for a clinical isolate in a polymicrobial infection would depend on the specific sensitive strains or species in the infection. If the burden is conferred by a plasmid-carried resistance gene, quantifying the burden would require the ability to cure plasmids, which remains a technical challenge[50,51]. However, the burden of plasmids from clinical isolates has limited variability even in new hosts and can be further reduced following compensatory mutations[52-54]. In contrast, the private benefit conferred by different Bla-expressing strains can vary widely depending on the enzyme and antibiotic uptake. Evolution experiments have been shown to produce striking changes in the single-cell MIC[55], which reflects the private benefit for a given cell, and different clinical isolates express widely variable resistance[18].

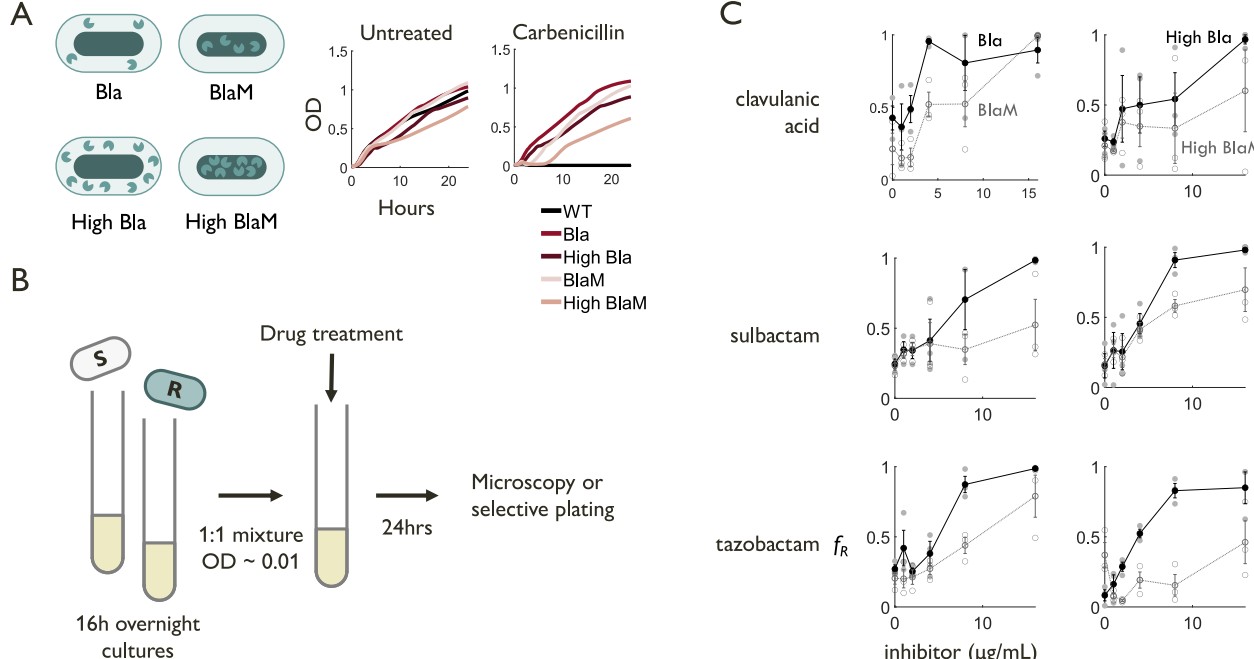

**Fig. 4 | Beta-lactam resistance is less selected for in strains with less private benefit. A** Plasmids used in the synthetic system include both Bla (periplasmic) and BlaM (cytoplasmic) variants expressed on vectors with either low (pBla, pBlaM) or high (pHSGBla, pHSGBlaM) copy number. Growth in the absence of carbenicillin shows that high-copy plasmids (dark red, dark pink) confer a higher burden (slower growth rate) than low-copy (light red, light pink) plasmids, which exhibit similar growth to plasmid-free (black, WT) strains. Growth in the presence of 100 μg/mL carbenicillin demonstrates that periplasmic variants (red) have a high private benefit and are unperturbed by the presence of antibiotic, while cytoplasmic variants (pink) have low private benefit and exhibit crash-and-recovery dynamics indicative of population-level rather than single-cell level antibiotic response. Average of 6 replicates reported. **B** Experimental flow for coculture experiments.

Resistant and sensitive strains are cultured overnight, then mixed in equal proportion and diluted to an estimated initial cell density of $10^7$ cells/mL in 1 mL of LB. The culture grows for 24 h in the presence of combination treatment before community composition is measured using fluorescence microscopy or selective plating. **C** Results from coculture experiments between plasmid-free sensitive and plasmid-carrying resistant strains in the presence of 2 μg/mL amoxicillin and increasing concentrations of clavulanic acid, sulbactam, and tazobactam ($n = 3$ biological replicates, shown in dark gray closed circles for Bla and light gray open circles for BlaM). All else being equal, the proportion of resistant cells $f_R$ is consistently higher for the higher private benefit Bla variant (black closed circles) than for the lower private benefit BlaM variant (medium gray open circles). Error bars represent the standard error of the mean.

Despite these challenges, the growth dynamics of clonal populations of different strains have a high information content that can be used for parameter estimation. In particular, private benefit and intracellular inhibition affect clonal responses of strains to β-lactam and combination treatments respectively, and can be quantified without genetic manipulation. To this end, we generated growth curves of a library of 311 clinical isolates in the absence of treatment and in response to 50 μg/mL amoxicillin and 50 μg/mL amoxicillin plus 25 μg/mL clavulanic acid (Fig. 5A, Supplementary Fig. S8)[24]. These isolates exhibited a wide spectrum of responses, including sensitivity to β-lactams alone; crash-and-recovery dynamics indicative of lower private benefit but collective antibiotic tolerance, or resilience[18], to β-lactams; resistance to β-lactams but susceptibility to β-lactam/Bla inhibitor combinations; and resistance to the combination treatment. These responses are quantitatively reflected by the distribution of growth rate perturbation and recovery times, which are the resistance and resilience metrics we previously formulated[18] (Fig. 5B).

To relate these growth curves to our model parameters and resulting predictions, we modified growth terms to better capture the experimental data and then used nonlinear optimization to estimate parameters for each of the 311 isolates by accounting for the growth of each strain under our three treatment conditions (Supplementary Figs. S6, S7). Experimental growth curves were well correlated ($R^2 = 0.948$, RMSE = 0.132) with the simulated curves generated using the estimated parameters (Fig. 5C, Supplementary Fig. S8). We find that our relative measure of private benefit, estimated $\beta_{min}$, exhibits a bimodal distribution (Fig. 5D, Supplementary Fig. S7), with ~80% having a $\beta_{min}$ below 0.5 (indicating a high private benefit). This is

consistent with the fact that most of these isolates are resistant or resilient in response to a β-lactam, amoxicillin. Moreover, estimated $\beta_{min}$ is correlated with resistance and resilience (Supplementary Fig. S9) and varied between strains (Fig. 5D, Supplementary Fig. S9). We note that whether these properties are entirely due to the expression of a Bla does not affect the predictive power of our model or the selection criterion (Eq. 1); the model predictions are based on the phenomenological measures of these parameters, regardless of underlying mechanisms.

Based on these estimates, we selected several resistant strains with varying values of estimated $\beta_{min}$ and generally slow or moderate growth (Supplementary Fig. S9). To investigate the selection dynamics of communities containing clinical isolates, we made mixtures of initially equal populations of our chosen clinical isolates and a β-lactam-sensitive laboratory strain, DA28102, and measured their composition after 24 h using selective plating. In general, our clinical isolates grow faster than DA28102 in the absence of antibiotic treatment. To better align with the simulation configuration, we use a sublethal concentration of chloramphenicol, to which DA28102 is resistant, to slightly suppress the growth rates of the clinical isolates. Under this condition, the clinical isolates grew slightly more slowly than DA28102 in the absence of β-lactam/Bla inhibitor combination treatment (Supplementary Fig. S10). In the presence of amoxicillin-clavulanic acid treatment, we found that using Bla inhibitors on strains with less private benefit (higher $\beta_{min}$) generally led to less resistant final compositions, while using Bla inhibitors on strains with greater private benefit (lower $\beta_{min}$) generally led to more resistant final compositions (Fig. 5E).

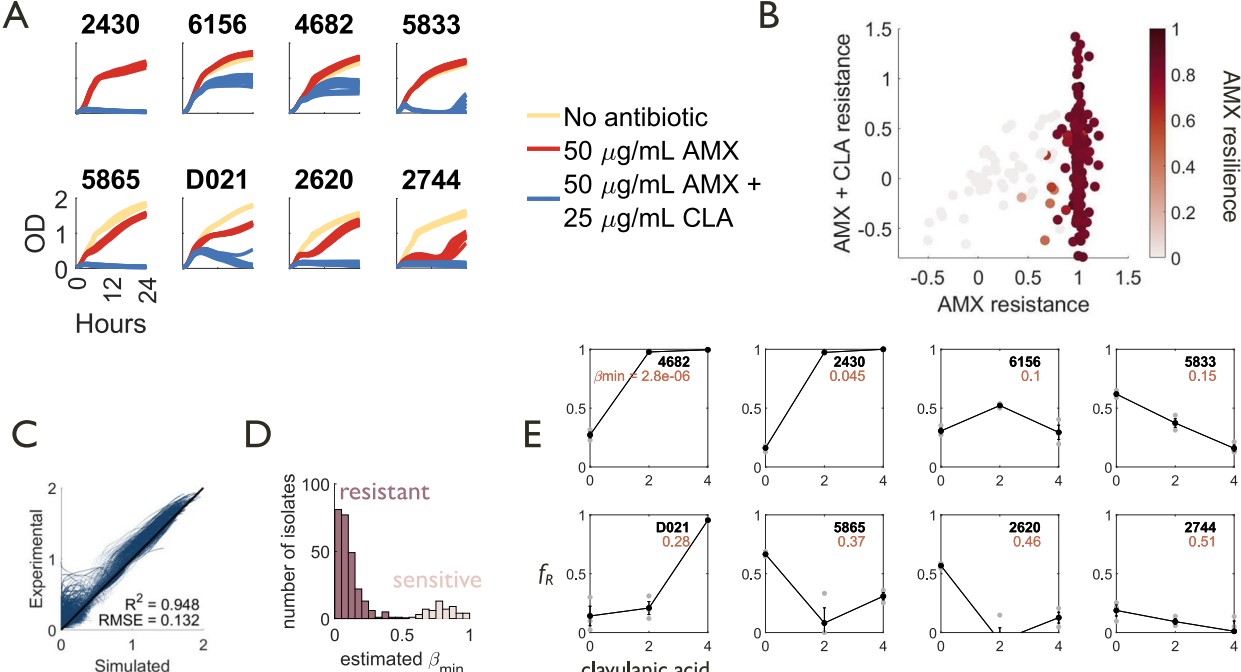

**Fig. 5 | Estimation and effects of private benefit in clinical isolates. A** Example subset of 11,196 growth curves generated for a library of 311 clinical *Enterobacteriaceae* isolates. Strains were grown under no antibiotic (beige), 50 μg/mL amoxicillin (AMX) (red), and 50 μg/mL amoxicillin with 25 μg/mL clavulanic acid (CLA) (blue) conditions ($n = 12$). A variety of responses to the three treatment conditions are observed, including responses that are characteristic of higher (no deviation from untreated condition) or lower (crash and recovery) private benefit. **B** Metrics of single-cell (resistance) and population-level (resilience) antibiotic tolerance show the distribution of responses for all 311 clinical isolates in the library. **C** Correlation between experimental data (average of 12 replicates) and simulated curves generated from estimated parameters after nonlinear optimization. Black line denotes $y = x$. $R^2 = 0.948$, root mean squared error RMSE = 0.132. When evaluating estimated parameters, we use the average value of each parameter from 10 rounds of optimization (see also Supplementary Figs. S6 and S7). **D** Distribution of estimated $\beta_{min}$ parameter, where $\beta_{min}$ is inversely correlated with private benefit. Strains separated into resistant (dark pink, recovered to at least 50% of maximum untreated population size within 24 h) and sensitive (light pink, never recovered to 50% of maximum untreated population size) subsets. **E** Coculture experiments between clinical isolates of varying $\beta_{min}$ values and a sensitive laboratory strain (DA28102) in the presence of 2 μg/mL amoxicillin and increasing concentrations of clavulanic acid confirm that strains with higher $\beta_{min}$ values can be selected against ($f_R$ decreases) when treated with β-lactam/Bla inhibitor combinations ($n = 3$, replicates shown in gray). Error bars represent the standard error of the mean.

To test whether our predictions hold in a more complex system than pairwise cocultures, we treated mixtures of our selected clinical isolates and a community of barcoded Keio strains which were sensitive to amoxicillin, where the densities of the resistant isolate and the mixed community of Keio strains were initially equal (Fig. 6). In the presence of amoxicillin-clavulanic acid treatment, we again found that using Bla inhibitors on strains with less private benefit (higher $\beta_{min}$) generally led to less resistant final compositions, while using Bla inhibitors on strains with more private benefit (lower $\beta_{min}$) generally led to more resistant final compositions (Fig. 6A). In order to verify that these selection dynamics occurred in a community context, we performed next-generation sequencing on the mixtures both before and after treatment. Our results show that community diversity was maintained during the course of treatment (Fig. 6B).

## Discussion

Our work provides a unifying framework to interpret and reconcile experimental observations of selection outcomes in response to β-lactam/Bla inhibitor combinations[20,22]. While different doses may result in selection for either sensitive or resistant populations for any given strain, our analysis reveals the key parameters that dictate the selective outcome. In particular, the function of Bla is often considered a public good, due to its ability to degrade β-lactams. Our results reveal the critical importance of the private benefit of Bla in dictating the outcome of selection dynamics in response to combination treatment (Eq. 1, Fig. 3). This insight is reminiscent of studies in other systems that demonstrate how privatization of a public good can ensure the maintenance of a cooperative trait[56,57]. While our analysis focuses on Bla-mediated resistance, the model framework and selection criterion are generally applicable to other types of antibiotics, where resistance can vary in the degree of sociality[58].

The private benefit of Bla is dictated by two factors (Eq. 1). One is the intrinsic private benefit, indicated by $\beta_{min}$; the other is the ability to suppress this private benefit with an inhibitor, indicated by $c$. Both parameters are dictated by specific combinations of bacterial strains and drugs. Our analysis of the clinical isolates reveals a wide diversity of these traits (Fig. 5, Supplementary Figs. S7, S9). Though they do not drastically affect the apparent overall drug response in terms of total cell densities (Fig. 3), they have profound impact on the underlying population structure at the end of treatment. Upon removal or turnover of the drugs, the residual populations may be more or less resistant, depending on the specific values of $\beta_{min}$ and $c$.

Therefore, our results suggest a need to measure or modulate such parameters. To this end, a critical insight from our analysis is that the effective private benefit varies among clinical isolates, but clonal growth measurements under different conditions can predict this property and the consequent selection dynamics of mixtures (Fig. 5). Although population composition experiments themselves can have low throughput, growth curves for parameter estimation can be done at high throughput. This estimation can be thought of as analogous to antibiotic susceptibility testing, where instead of predicting whether the population is susceptible to an antibiotic, it predicts whether it is possible to select against resistance with a combination.

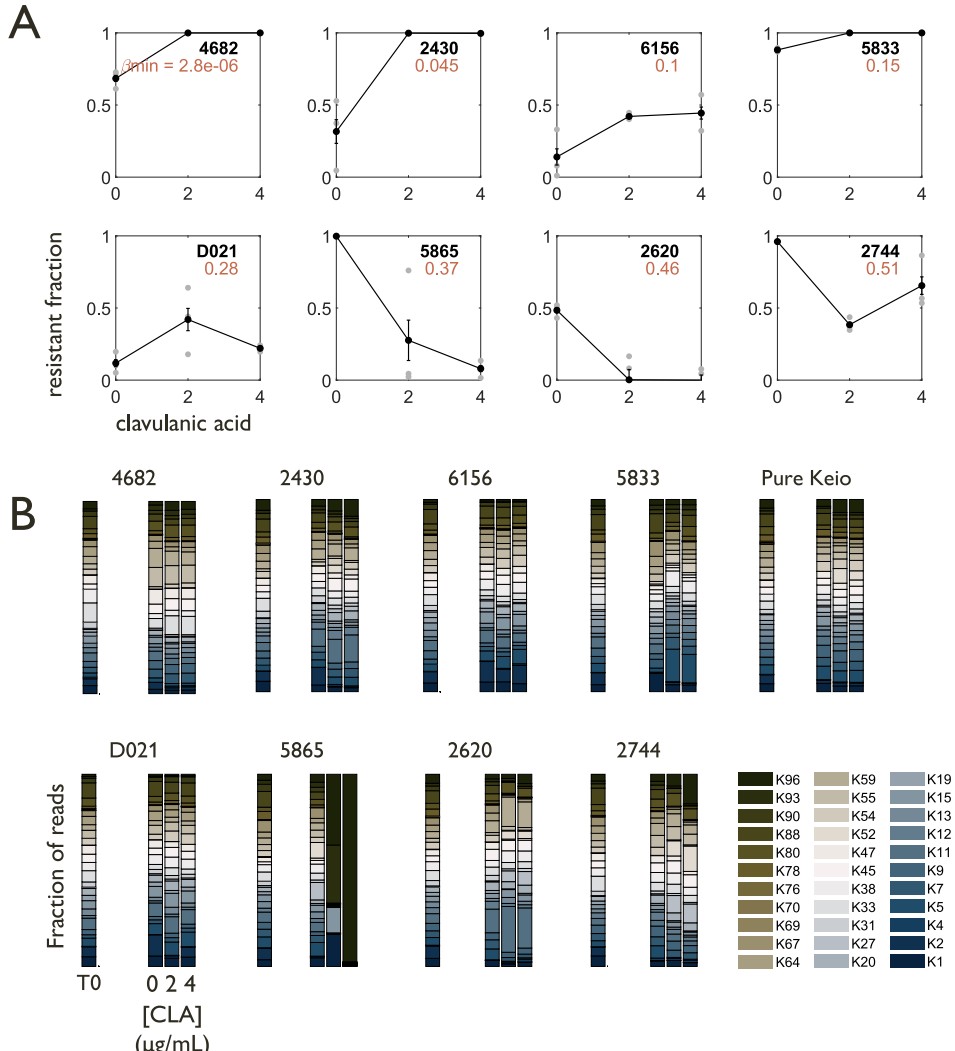

**Fig. 6 | Estimated private benefit is associated with selection outcomes for resistant cells cocultured with sensitive communities.** Cocultures initialized with equal initial densities of a clinical isolate and a community of barcoded sensitive Keio strains (Supplementary Table S1) were treated with 2 µg/mL amoxicillin and increasing concentrations of clavulanic acid. The initial and final mixtures were sampled for cfu plating and barcode sequencing. **A** Resistant strains with higher $\beta_{min}$ values can be selected against when treated with β-lactam/Bla inhibitor combinations, even when embedded in a broad community of sensitive strains rather than in a pairwise mixture. Replicates shown in gray ($n = 3$ biological replicates).

Error bars represent the standard error of the mean. **B** For almost all treatment-resistant strain combinations, the sensitive community did not become dominated by any one strain, verifying that the selection dynamics of the resistant strain were indeed occurring in a sensitive community context. The only exception was the 5865-Keio mixture, which became dominated by 4 (K1, K15, K93, K96), and 1 (K96) strains, respectively, when treated with 2 and 4 µg/mL clavulanic acid (CLA). Stacked bars indicate the fraction of total reads aligned to each strain's barcode as a fraction of the total, presented as the average of 3 biological replicates.

When available, these parameters can guide choice of treatment strategies. For example, for some clinical isolates (with large $\beta_{min}$), the lowest resistant fractions were observed at high Bla inhibitor concentrations (Figs. 5E, 6A). The historical development of drug combinations like amoxicillin/clavulanate, in which the relative concentration of antibiotic has been increased from 2:1 to formulations as high as 16:1, has been driven by the recognition that the time above MIC is the key parameter that determines treatment efficacy[59,60]. While increasing antibiotic concentration increases the time above the MIC, so does increasing inhibitor concentration, by lowering the antibiotic's MIC rather than increasing its concentration. In the long term, giving clinicians more freedom to set separate doses of antibiotic and inhibitor components may facilitate dose optimization that reduces resistance.

Our analysis also highlights the importance of the intracellular inhibition of Bla. Current guidelines do not differentiate between combinations with different Bla inhibitors: for instance, amoxicillin/

clavulanate and ampicillin/sulbactam, which have β-lactam components with similar spectra of activity, may be used interchangeably. However, our results suggest the need to consider inhibitor permeability, which depends on both the target bacteria and the specific inhibitor[28,29], when choosing between otherwise equivalent therapies. In recalcitrant infections, the pairing of more effective antibiotics with antibiotic-inhibitor combinations with more permeable inhibitors can even be considered. This unorthodox method of obtaining different antibiotic-inhibitor pairs has been successful in some cases as salvage therapy[61,62], and suggests that regulatory approval of combinations which partner the most permeable of existing inhibitors with different antibiotics could provide more avenues to successful treatment. Adjuvants that increase inhibitor penetration should also be developed. Permeabilizing adjuvants have already been shown to boost the activity of various antibiotics[63] and even Bla inhibitors[64]; our work suggests that they may boost the resistance-suppressing ability of these inhibitors as well.

Decision-making in the clinic is frequently empirical and difficult. Paired with improvements in fast diagnostics at the strain level, databases of strains which record their key parameters will enable the development of algorithms that can make optimal treatment recommendations at the time of diagnosis, rather than after a lengthy period during which the pathogen is characterized. This increasingly personalized approach will ultimately enable more evidence-based clinical decisions that can account for both the needs of the individual patient and society's need to combat antibiotic resistance.

## Methods

### Bacterial strains and plasmids

The low copy Bla and BlaM-expressing plasmids for our synthetic strains, pBla-sfGFP and pBlaM-sfGFP, were constructed by inserting superfolder GFP into the pBla and pBlaM plasmids[23,65] under the constitutive promoter J23115. The pBla plasmid is derived from the pPROlar.A122 (Clontech) vector, with a p15A origin of replication, expressing kanamycin resistance constitutively and expressing *bla* under the control of the $P_{lac/ara-1}$ promoter. The high copy Bla and BlaM-expressing plasmids for these strains, pHSGBla-sfGFP and pHSGBlaM-sfGFP, were constructed by Gibson Assembly from the pHSG298 plasmid (Takara Bio), with a pMB-derived origin of replication and expressing kanamycin resistance constitutively. Bla and BlaM were inserted into the multiple cloning site under a *lac* promoter and sfGFP was inserted under J23115. All plasmids were transformed into DA28102 (K-12 F− λ− *ilvG−rfb-50 rph-1 galK::cat*-J23101-mTagBFP2)[66] or MC4100Z1 (K-12 *araD139 (argF-lac)205 flb-5301 pstF25 rpsL150 deoC1 relA1*, with chromosomally inserted Z1 cassette containing *lacIq, tetR*, and *spect(R)* genes, gift of M. Elowitz).

Our library of 311 clinical isolates was sourced from the Duke University hospital (supplied by Vance Fowler and Joshua Thaden) and North Carolina community hospitals (supplied by Deverick Anderson)[67]. Further genomic and phenotypic analysis of some of these strains and their plasmids can be found in our previous work[41,44,54]. The library consists primarily of *Escherichia coli*, many expressing ESBLs and multidrug-resistant, with a minority of strains from *Klebsiella, Citrobacter*, and *Enterobacter* genera. No clinical information regarding the clinical isolates was used in the study. Laboratory operations on these isolates follow protocols approved by the Duke University Health System Institutional Review Board.

Our library of Keio strains, with single-gene knockouts replaced by kanamycin cassette and carrying a plasmid containing unique barcodes and chloramphenicol resistance, was previously constructed by our group[68].

### Growth media

All experiments in this study were performed in LB Broth, and all plates were made with LB agar (Miller, Apex Bioresearch). Unless stated otherwise, all overnight cultures were started in 16 mL culture tubes in 2 mL LB from colonies on LB agar with selection. If carrying a synthetic plasmid, 1 mM IPTG and 50 μg/mL kanamycin were added to maintain selection pressure and induce beta-lactamase production. All overnight cultures were grown at 37 °C for 16 h, with shaking at 225 rpm. For each experiment, solutions of amoxicillin (Sigma) or carbenicillin (ThermoFisher) and the beta-lactamase inhibitors (clavulanic acid, Sigma; tazobactam, Fisher Scientific; sulbactam, Fisher Scientific) were prepared fresh in DMSO (amoxicillin, tazobactam) or water (carbenicillin, clavulanic acid, sulbactam).

### Dose-response matrices

Overnight cultures were corrected to an OD600 of 1. For two-population experiments plasmid-free and plasmid-carrying cells, respectively, were mixed in equal proportion. Corrected overnight culture or two-population mixture was diluted 1:16 for an assumed density of $5 \times 10^7$ cells/mL, and 50 μL of 1 M IPTG was added to induce the beta-lactamase enzyme. Stock solutions of amoxicillin and the respective Bla inhibitor were diluted into LB at concentrations 2.5 times the final concentrations. Using a MANTIS® liquid handler (Formulatrix), 40 μL of the appropriate antibiotic and inhibitor-containing solutions were dispensed into each well in a 384-well deep well plate (Thermo Scientific), followed by 20 μL of the diluted culture. Final initial cell density was $1 \times 10^6$ cells per 100 μL well, final IPTG concentration was 1 mM, and final antibiotic and inhibitor concentrations formed a dose-response matrix of 0, 0.5, 1, 2, 4, 8, 16, 32, 64, and 128 μg/mL of the agents. Three replicate wells corresponded to each condition, and well positions were randomized across the plate, with blank LB dispensed into all edge wells and 12 interior wells. To minimize evaporation, the plate was loaded with lid into the plate reader (Tecan Spark), which was equipped with a lid lifter, and the chamber temperature was maintained at 30 °C. OD600 and GFP readings were taken every 10 min with periodic shaking (5 s orbital) for 24 h.

For Fig. 1A, experiments were conducted in a 1536-well plate (Greiner Bio-One). The procedure was the same as above with the following modifications: Stock solutions of cefotaxime were diluted into LB at 2000 and 80 μg/mL and stock solutions of clavulanic acid were prepared at 20 and 0.8 μg/mL. They were dispensed into wells for final concentrations of 0, 0.8, 1.6, 2.4, 4, 5.6, 9.6, 14.4, 20, 40, 60, 80, 140, 200, 320, and 500 μg/mL (cefotaxime) and 0, 0.048, 0.072, 0.096, 0.136, 0.184, 0.2, 0.4, 0.6, 0.8, 1, 1.4, 1.8, 2.6, 3.6, and 5 μg/mL (clavulanic acid). These values were chosen to increase in log scale between the minimum and maximum concentrations but were rounded off to account for the limitations of the liquid handler. Final cell density was a 1:1000 dilution from overnight culture. Each condition had five replicate wells. OD600 readings were taken every 20 min with periodic shaking (5 s orbital) for 40 h.

### Modeling selection dynamics

We formulated ordinary differential equations to model the dynamics of a bacterial system with resistant and sensitive subpopulations responding to a β-lactam/Bla inhibitor combination. The model describes interactions between five main components: sensitive population density ($n_s$), resistant population density ($n_r$), nutrient level ($s$), antibiotic concentration ($a$), and Bla concentration ($b$). Key processes include the costs ($\alpha$) and benefits ($\beta$) of Bla production, antibiotic degradation by living resistant cells ($\varphi$), the effect of Bla inhibitor ($i$) on Bla activity ($d_b$) and private benefit (modulated by $c$), nutrient recycling ($\xi$), antibiotic degradation by Bla ($\kappa_b$), and basal antibiotic degradation ($d_a$). The effects of antibiotic and inhibitor are both saturating according to a Hill equation with Hill coefficients ($h_a$) and ($h_i$) respectively, which reflect the steepness of the dose-response curve. Our model equations are as follows:

$$\frac{dn_s}{d\tau} = (g - l)n_s$$

$$\frac{dn_r}{d\tau} = (\alpha g - \beta l)n_r$$

$$\frac{ds}{d\tau} = (\xi l - g)n_s + (\xi \beta l - \alpha g)n_r$$

$$\frac{da}{d\tau} = -\kappa_b ba - \varphi n_r a - d_a a$$

$$\frac{db}{d\tau} = \beta l n_r - d_b i b$$

$$g = \frac{s}{1+s}$$

$$l = \gamma \frac{a^{h_a}}{1+a^{h_a}} g$$

$$\iota = \frac{i^{h_i}}{1+i^{h_i}}$$

$$\beta = \beta_{min} + c(1-\beta_{min})\iota$$

$$\varphi = \varphi_{max}(1-c\iota)$$

All simulations were conducted in MATLAB R2019b and R2023b. Further details of model formulation assumptions, initial conditions, criterion derivations, and parameter ranges used in modeling results can be found in the Supplement.

### Population composition assays for synthetic system
To test the population dynamics of sensitive and resistant strains under beta-lactam/Bla inhibitor treatment, we corrected overnight cultures to OD600 of 1 and mixed plasmid-free and plasmid-containing in equal proportion. We used the pre-diluted mixture to measure our initial composition. We then diluted the mixture 80-fold in 1 mL of LB in a 16 mL culture tube, with 2 µg/mL amoxicillin and appropriate Bla inhibitor concentrations for a starting density of $10^7$ cells/mL. Cultures were grown for 24 h at 37 °C with shaking at 225 rpm, and final compositions and OD600s were measured. Independent overnight cultures from different bacterial colonies were used for different replicates.

To measure population composition, we utilized the fluorescent markers on DA28102 and the plasmids. For each replicate, we sampled 1.5 µL of each culture on a slide with 0.15 mm cover glass and, at three different points on the slide, took z-stacks (5 images at 0.9 µm difference, to account for variations in focus in the field of vision) using phase contrast, DAPI or DAPI-V, and GFP filters with a 40X objective on the Keyence BZ-X710 or BZ-X800. We integrated z-stacks by taking the darkest value from phase contrast and the mean value from fluorescent channels. We then applied thresholds to each image, which were set to be inclusive for fluorescent channels and discriminative for phase contrast. Pixels were considered to belong to living cells if they were both dark in phase contrast and either green or blue. Pixels were considered to belong to resistant cells if they were in the above set and were green. Biomass for total living and resistant cells was determined by summing areas across all three images and the resistant fraction was then determined by dividing resistant biomass by living biomass.

### High-throughput growth phenotyping of clinical isolates
The generation of the antibiotic-free experimental data used in this work was described in Zhang et al.[41]. For this dataset, we used an augmented library of 311 clinical Enterobacteriaceae isolates collected from the Duke University Hospital and North Carolina community hospitals. Three separate colonies were selected to inoculate growth media. Overnight cultures were prepared in 1 mL of LB broth in 96-well deep-well microplates (VWR), which were shaken at 37 °C for 16 h at 1000 rpm with a BreatheEasy film (USA Scientific). The OD600 for the overnight culture was taken on a plate reader (Tecan Spark). Cultures were diluted to OD600 1 and further diluted 1:8 ($1 \times 10^8$ cells/mL). Cultures were then finally diluted 10-fold in 100 µL of fresh media in a 384-well deep-well plate (Thermo Scientific) using a MANTIS liquid handler for an initial cell density of $1 \times 10^6$ cells/well. LB media was used for all experiments, with three culture conditions:

(1) no antibiotic treatment, (2) 50 µg/mL amoxicillin, and (3) 50 µg/mL amoxicillin + 25 µg/mL clavulanic acid.

Each overnight culture (three biological replicates) was used to inoculate four wells for each condition (to generate four technical replicates), for a total of 12 replicates. The spatial position of all wells for each experiment was randomized across the plate to minimize plate effects, with blank LB dispensed into all edge wells. The plate was loaded with the lid into the plate reader, and the chamber temperature was maintained at 30 °C. OD600 readings were taken every 10 min with periodic shaking (5 s orbital) for 24 h.

### Nonlinear optimization of model parameters
To better account for features of our experimental data, we modified the growth term of our model and introduced a cell density threshold to trigger the beginning of lysis. We used the untreated condition to optimize for growth-specific parameters, then fixed the results (and seven more parameters) during optimization. We ran ten rounds of optimization using the scipy.optimize implementation of the Nelder-Mead algorithm, minimizing the difference between the simulated time courses and the average of 12 replicates for all three experimental conditions for each isolate. Initial guesses for each round were taken from a Gaussian distribution within each parameter range. Full details can be found in Supplementary Information.

### Population composition assays for clinical isolates with a single sensitive strain
To test how strains with different estimated $\beta_{min}$ would be affected by β-lactam/Bla inhibitor combinations, we chose clinical isolates with a variety of estimated $\beta_{min}$ and low estimated $\alpha$ (low growth rate) where possible (Supplementary Fig. S9). Setup for population composition assays for clinical isolates were done analogously as to the synthetic system. However, we added 5 µg/mL chloramphenicol (Sigma) to ensure that our beta-lactam-sensitive strain, DA28102, would grow faster than the clinical isolates (Supplementary Fig. S10). Because the clinical isolates do not express fluorescence, we used selective plating to distinguish chloramphenicol-expressing (DA28102, β-lactam sensitive) cfus from total (DA28102 and isolate combined) cfus. Cfus on each type of plate were calculated as the average from three replicate plates. Resistant fractions were calculated as $\frac{cfu_{LB} - cfu_{LB+cm}}{cfu_{LB}}$.

### Population composition assays for clinical isolates with a sensitive Keio community
Setup for assays with the sensitive Keio community were done analogously to those with the single sensitive strain with the following differences. Rather than coming from a single overnight culture of the sensitive strain, overnight cultures of each Keio strain, prepared in 0.5 mL of LB broth in 96-well deep-well microplates shaken at 37 °C for 16 h at 700 rpm with a BreatheEasy film, were mixed in equal proportion and then corrected to OD600 of 1 as a whole before mixing in equal proportion with the resistant isolate. The assay was also conducted in a 96-well deep-well microplate shaken at 37 °C for 24 h at 700 rpm with a BreatheEasy film. Cfus from experimental wells on each type of plate were calculated as the average of two replicate plates, and three separate wells for each condition were used as replicates. The list of strains can be found in Supplementary Table ST1.

### Keio community barcode counts
200 µL of the initial Keio/isolate mixture and each treatment well after 24 h of growth were sampled and centrifuged at 2000 g for 2 min. Supernatant was discarded and the pellet was resuspended in 20 µL PBS, boiled at 95 °C for 5 min, and stored at 4 °C. Prior to sequencing, we diluted samples by between 1:2 and 1:8 as necessary based on cfu counts on chloramphenicol plates to even out expected DNA amounts.

To prepare the next-generation sequencing library, we followed previously established protocols[68,69]. In brief, we ligated dual indexes,

sequencing primers, and adapters using a 2-cycle PCR, using the following forward and reverse primers for the first step: CCGACCACCG AGATCTACACXXXXXXXXXYYYYYYYYYYAACACTCTTTCCCTACACGA CG,CAAGCAGAAGACGGCATACGAGATXXXXXXXXXGTGACTGGAGTT CAGACGTGTGC. Here Ys represent unique molecular identifiers and Xs represent dual-sample indexes. DNA was denatured for 30 s at 98 °C, followed by two PCR cycles of 10 s at 98 °C, 30 s at 67 °C, 20 s at 72 °C and a final extension for 5 min at 72 °C. We then carried out 2-step size selection using SPRIselect magnetic beads (Beckman Coulter). Sequences were pooled and amplified using the following forward and reverse primers: AATGATACGGCGACCACCGAGATCTAC, CAAGCAGA AGACGGCATACGAG. DNA was denatured for 30 s at 98 °C, followed by 30 PCR cycles of 10 s at 98 °C, 30 s at 66 °C, 20 s at 72 °C and a final extension for 5 min at 72 °C. Products in the 190-290 bp range were purified through gel extraction (Zymoclean Gel Recovery). We then prepared the final library according to manufacturer guidelines for the Illumina MiniSeq platform, including a 30% PhiX spike-in. We used the Miniseq and its software to perform paired-end sequencing.

We aligned the resulting fastq files using a Galaxy workflow using FastQC (https://www.bioinformatics.babraham.ac.uk/projects/fastqc/), FastQ joiner[70], Barcode Splitter (http://hannonlab.cshl.edu/fastx_toolkit/), Trim Sequences (http://hannonlab.cshl.edu/fastx_toolkit/), and FLASH[71]. We used a custom Python script to count aligned reads with up to 2 bp of mismatch. Finally, we calculated fraction for each strain for each condition by dividing the count for each Keio strain by the total counts for all Keio strains for that condition. Samples from each of three separate wells for each condition were used as replicates.

### Reporting summary

Further information on research design is available in the Nature Portfolio Reporting Summary linked to this article.

## Data availability

Source data are provided with this paper as a Source Data File. All data is also deposited at: https://github.com/youlab/combination-antibiotic-selection[72], and we recommend using this analysis infrastructure to directly access data for specific figures, including Supplementary Figs. Source data are provided with this paper.

## Code availability

Code is available at: https://github.com/youlab/combination-antibiotic-selection.

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

## Acknowledgements
We thank Vance Fowler and Joshua Thaden for contributing isolates. Scientific color maps from the cmocean (Thyng et al. 2016) and fcrameri (Crameri 2018) packages are used to minimize visual distortion and exclusion of readers with color-vision deficiencies. This work was partially supported by the National Institutes of Health (L.Y.: R01AI125604, R01GM098642; L.Y. and D.J.A.: R01EB031869), US-Israel Binational Science Foundation (L.Y. 2021192), and an NSF graduate fellowship (H.R.M.). The funders had no role in study design, data collection and analysis, decision to publish, or preparation of the manuscript.

## Author contributions
H.R.M. and L.Y. conceived the research and designed the research framework, with input from D.J.A. H.R.M. and K.K. conducted experimental analysis with assistance from H.X. H.R.M. conducted modeling analysis. D.J.A. provided the clinical isolates. H.R.M. and L.Y. wrote the paper with inputs from H.Z.X., K.K., and D.J.A.

## Competing interests
The authors declare no competing interests.
