## [Peer Review File · Nature Communications]

REVIEWER COMMENTS

Reviewer #1 (Remarks to the Author):

The manuscript combines a mathematical modelling with experimental verification of their model predictions for selection for resistance with a combination application of antibiotics and beta-lactamase inhibitors.

The theoretical results include a simple inequality to determine the competitive advantage (higher growth rate) of the resistant strain. This theoretical result is tested against synthetic strains and clinical isolates. The authors claim that their results can easily be adopted because the parameters required can easily be obtained from single population growth experiments.

This work is relevant to inform drug (combination) design, but also to the more fundamental fields of microbial population dynamics.

I have three major concerns in the results and interpretation of the study.

1. In Figure 3C there seems to be a discrepancy between expected theoretical outcome (black line) and the simulations (blue-red dots). This discrepancy seems to increase with higher values of the two side of the inequality (as plotted on the x-axis and y -axis). This is not explained in the results or discussion. The authors actually say that it is an "excellent predictor". I would agree if this would have just be random noise around the boundary, but it seems to be systematic.

2. Genes coding for Beta-lactamases are often found on plasmids. The authors recognize this. The model does, however, not include any horizontal gene transfer (HGT) for example by conjugation. This choice is not clearly substantiated and might actually change the inequality. If HGT plays a role in the populations. This reduces the applicability of the theoretical outcome. The authors needs at least mention this limitation of their approach.

3. Generalizability to other environments (e.g. within-patient) is less than shown in this paper. The authors did not prove that there results would also hold in more complex communities (e.g. three populations). It is known that such communities can express complex behaviour, which might result in different outcomes. Although the authors do mention that in the case that rest of the community might be seen as the sensitive strain, this is a big assumption and there is no evidence from their experiments that this would work like that.

Minor:

- Page 7: - Why only look at majority vs minority. Isn't it more interesting to see if there is coexistence yes/no? Even a minor population of resistant bacteria can cause a problem.

The authors indeed make this point in the next paragraph. I actually disagree that the 'evolutionary objective' is a not part of antibiotic stewardship. I think this is included in the definition by e.g. the CDC.

- page 8: The introduction of parameter c is vague. For a modeller it is not clear what this parameter will do (except if you dig into the supplement). The parameter is also not in the explanatory diagram (figure 2A)

- Page 11. Why does the bactericidal dose of inhibitor increase selection for resistance. I would think that this effect is equal for resistant and sensitive strain.

- Page 13. It is not written which correlation coefficient is reported. More importantly, I do not think the Pearson's correlation coefficient is a good Goodness-of-Fit measure. If it would be than $R^{2\text{sup}}$ would be better,

but it is not in the case of non-linear curve fitting. GoF for non-linear curve fitting is non trivial, I think this requires a statistician (which I am not). Just looking at it (eye-ball test so to say) tells me that the fit is okay though, that is why this is a minor issue in the paper.

- Figure 3A: unit misses in the scale legend

Supplementary information:

Major:

- Line 31-32: ",the additional complexity does not change the qualitative aspects of the our conclusions".

It is not clear how the statement is substantiated. Is this based on literature or did the authors actually test this.

- Line 33-34 Did the authors try simulations with different initial values? And if so did that change the outcomes?

Minor:

First page.

- In the description of the model the description and preferably interpretation of the Hill coefficient (h_a and h_i) are missing.

- Monod constant could be given as value or with a reference.

Throughout:

- use equation editor also in-line such that the symbols in the equations match the once in text.

Currently hard to tell the difference between l ('el') and ι ('iota').

Reviewer #2 (Remarks to the Author):

Comment:

The goal of the study is difficult to define and has not been clearly stated. However the study reveals certain interesting points:

- Variation in effective treatment doses and complex growth dynamics for mixed sensitive and resistant populations which is very interesting.
- Combination treatment impacts selection dynamics based on strain and drug-specific factors, including Bla production's private benefit, inhibitor suppression, and burden.
- Quantifying phenotypic bacterial responses for effective treatment strategies against resistant strains.

The author should define clear objective as well as importance of this study in terms of tackling resistance.

Reviewer #3 (Remarks to the Author):

This is a very interesting article that brings together experimental and modelling work on beta lactamase

action, and in particular in the context of private benefit of periplasmic beta lactamase genes vs "public" benefit of cytoplasmic beta lactamase genes that become available to other cells following cell lysis. This is explored in the context of combined treatment with beta lactams and beta lactamase inhibitors.

While the paper is very interesting, I have serious reservations about the structure of the model and the underlying hypothesis of the study. The model considers a single population of resistant cells and single type of beta lactamase. However, the paper has at its core the comparison between strains that produce differing amounts of cytoplasmic beta lactamase enzymes (which then enter the extracellular space following cell lysis) and periplasmic beta lactamase enzymes (which are only available to producer cells and presumably do not enter the extracellular space following cell lysis). The normal ecological argument that would be made in such a system is that there would be a balance between strains that produce a public good and those that consume it without producing it – i.e. ‘cheater’ strains that benefit from the cytoplasmic beta lactamase enzyme produced by other cells but which do not bear the cost of its production. Of course this system is both more interesting and complicated because the public good is only available after the lysis of the producer cells, leading to potential also for kin selection dynamics. The model seems to describe none of these dynamics. At the very least I would expect to see two different beta lactamase genes in the model: the cytoplasmic and periplasmic genes each having their own equations and dynamics: periplasmic beta lactamase only benefitting the producer cells, while cytoplasmic beta lactamase benefitting the producer cells through internal concentration and all cells through external concentration. This is pretty much essential for meaningful modelling of the system and appears to be missing from the model – in fact the model formulation does not appear to be a good description of either of these beta lactamase genes. The model then needs to be adapted so that it is clear how it could describe (i) cells that only produce cytoplasmic bla (ii) cells that only produce periplasmic bla (iii) cells that produce both (iv) ideally also (from an ecological point of view) cells that produce neither but which could benefit from cytoplasmic bla from other cells.

As currently formulated, the model in the paper doesn't appear to be able to describe any of the cells in the paper. And as soon as it is thought through on these terms, it becomes clear that single population dynamics are insufficient to decouple private vs public goods advantages of the system. For example, cells that only produce periplasmic bla would benefit from their neighbours who produce cytoplasmic bla but not vice versa. One of the advantages of the model is that it could be used to examine mixed populations. The biggest weakness of the paper is that it is trying to understand a complex ecological idea from single strains – and this impacts both on the experimental and modelling work.

Other specific comments:

Abstract:

We note that both the title and abstract highlights the importance of the “private” benefit of bla production. Implicit in this statement is that there may be a contrasting public “benefit” argument (or even public goods game problem) that frames this private benefit. The Abstract needs the public context in order for that private argument to be made. Please can a relevant sentence be added in order to ensure correct logical flow.

Results

L85-125

This section is somewhat confusing because it mixes material that belongs in the Introduction (including reference to previous work, and material on private vs public benefits of Bla proteins) with results produced in this paper. This needs to be remedied. Relevant introductory material needs to be moved to the Introduction and the Results should ONLY describe the experimental results from this study. The Introduction would also massively benefit e.g. by discussion of periplasmic vs cytoplasmic Bla enzymes, as well as some further description of the likely consequences of private vs public goods in this context (to benefit those readers who might have a more molecular rather than ecological understanding of drug resistance).

L98 statement is vague. Can you specify in what way the responses are “profound”? It is actually difficult to understand Figure 1a on the basis of what is written both in the Results text and the figure caption so some more explicit explanation for the reader is needed.

L112-114. The reader does not know which strains have more private benefit: there is insufficient description of the strains in the Results to be able to clearly follow the argument. I appreciate that it is sometimes harder to write papers when Methods are at the end (rather than before Results) but some brief synopsis of what strains are involved is needed in the Results section to make the Results interpretable.

Point-by-point responses to reviewers' comments

REVIEWER COMMENTS

Reviewer #1 (Remarks to the Author):

The manuscript combines a mathematical modelling with experimental verification of their model predictions for selection for resistance with a combination application of antibiotics and beta-lactamase inhibitors.

The theoretical results include a simple inequality to determine the competitive advantage (higher growth rate) of the resistant strain. This theoretical result is tested against synthetic strains and clinical isolates. The authors claim that their results can easily be adopted because the parameters required can easily be obtained from single population growth experiments.

This work is relevant to inform drug (combination) design, but also to the more fundamental fields of microbial population dynamics.

I have three major concerns in the results and interpretation of the study.

We thank the reviewer for their recognition of how our work may inform drug dosing and our understanding of population dynamics. Below, we address their concerns with corresponding revisions to the manuscript.

1. In Figure 3C there seems to be a discrepancy between expected theoretical outcome (black line) and the simulations (blue-red dots). This discrepancy seems to increase with higher values of the two side of the inequality (as plotted on the x-axis and y -axis). This is not explained in the results or discussion. The authors actually say that it is an "excellent predictor". I would agree if this would have just be random noise around the boundary, but it seems to be systematic.

The reviewer raises a valid point. There is a deviation, which likely arises from the assumptions made when deriving the criterion. For example, our criterion considers only initial dynamics, which does not account for how antibiotic concentrations decrease over the course of treatment, allowing the growth advantage of sensitive populations to dominate. Considering the strength of the assumptions, the criterion is still a good predictor of final community composition. However, we have added text that clarifies how the discrepancy may arise. We have also highlighted that the discrepancy is in the direction of more conservative assumptions. That is, the predictor will err on the side of assuming something cannot be treated with this combination in a resistance-minimizing way when it actually can. The converse is more risky (assuming something can be treated but inadvertently increasing resistance).

2. Genes coding for Beta-lactamases are often found on plasmids. The authors recognize this. The model does, however, not include any horizontal gene transfer (HGT) for example by conjugation. This choice is not clearly substantiated and might actually change the inequality. If HGT plays a role in the populations. This reduces the applicability of the theoretical outcome. The authors needs at least mention this limitation of their approach.

We thank the reviewer for their insightful comment. Indeed, horizontal gene transfer is an important modulator of community structure that is not accounted for in our existing model. To avoid this confounding factor, we used isolates carrying nonconjugative plasmids in the experimental work.

In light of the reviewer's comment, we have carried out additional analysis and derived a criterion that accounts for the potential contribution of conjugation. If horizontal gene transfer is present, the forms of the cell density equations in the model become:

$$\frac{dn_s}{d\tau} = (g - l)n_s - \eta n_r n_s + \sigma n_r \quad S14$$

$$\frac{dn_r}{d\tau} = (\alpha g - \beta l)n_r + \eta n_r n_s - \sigma n_r \quad S15$$

where η is the plasmid transfer rate and σ is the plasmid loss rate. In this case, using the same assumptions as specified above, the criterion for enriching resistant cells becomes:

$$(1 - c)(1 - \beta_{min}) + \frac{\eta(n_{r0} + n_{s0})}{\gamma g_0} > \frac{1 - \alpha}{\gamma} + \frac{\sigma(n_{r0} + n_{s0})}{n_{s0} \gamma g_0}$$

Therefore, when conjugation is present, plasmid transfer rates higher than loss rates favor selection for resistant cells. This result highlights the importance of developing engineering solutions to minimize transfer and maximize loss.

We have added a paragraph presenting how the inclusion of conjugation and segregation error affect the criterion to the Results, with more details in the Supplemental Information.

3. Generalizability to other environments (e.g. within-patient) is less than shown in this paper. The authors did not prove that their results would also hold in more complex communities (e.g. three populations). It is known that such communities can express complex behaviour, which might result in different outcomes. Although the authors do mention that in the case that rest of the community might be seen as the sensitive strain, this is a big assumption and there is no evidence from their experiments that this would work like that.

The reviewer raised a valid limitation of our study, which we had limited to two-population communities. To address this limitation, we have conducted extensive additional experiments in which each of the resistant isolates in Figure 5 were cocultured with a mixture of 33 sensitive strains. The results are presented in Figure 6. In these experiments, the dynamics between the resistant strain and the sensitive community as a whole yield similar results as in Figure 5E: selection is greater against resistant strains with higher β_{min} (i.e., the Bla is more of a public good).

We used NGS to confirm that the community did not become dominated by any one strain throughout the course of treatment.

Minor:

- Page 7: - Why only look at majority vs minority. Isn't it more interesting to see if there is coexistence yes/no? Even a minor population of resistant bacteria can cause a problem.

We agree with the reviewer that small resistant subpopulations may lead to adverse outcomes. We focused on majority vs minority in this case because our experiments were initialized with an equal mixture of the two subpopulations. The majority/minority distinction thus indicates the direction of selection, and shows whether it is possible to select against the resistant subpopulation at all. We looked only at the effects of a single dose: longer term treatment, through repeated dosing, can continue selection to drive resistant populations to extinction. We have added a sentence to specify this to the paragraph mentioned in the next comment.

The authors indeed make this point in the next paragraph. I actually disagree that the 'evolutionary objective' is a not part of antibiotic stewardship. I think this is included in the definition by e.g. the CDC.

We thank the reviewer for pointing this out and have revised the text to clarify and acknowledge that the overall goal of antibiotic stewardship does include combating antibiotic resistance.

- page 8: The introduction of parameter c is vague. For a modeller it is not clear what this parameter will do (except if you dig into the supplement). The parameter is also not in the explanatory diagram (figure 2A)

We thank the reviewer for pointing this out and agree that we need to better introduce this key parameter. We have limited Figure 2A to state variables for simplicity. However, we have added some sentences to the previous section where the logic of the model is reduced that denotes the corresponding parameters and provides intuition for how they affect the relevant processes. We hope that this provides additional clarity for readers.

- Page 11. Why does the bactericidal dose of inhibitor increase selection for resistance. I would think that this effect is equal for resistant and sensitive strain.

We thank the reviewer for noticing this. They are correct that this is not a fair assumption, and we have removed the relevant sentences.

- Page 13. It is not written which correlation coefficient is reported. More importantly, I do not think the Pearson's correlation coefficient is a good Goodness-of-Fit measure. If it would be than R^2 would be better, but it is not in the case of non-linear curve fitting. GoF for non-linear curve fitting is non trivial, I think this requires a statistician (which I am not). Just looking at it (eye-ball test so to say) tells me that the fit is okay though, that is why this is a minor issue in the paper.

We thank the reviewer for pointing out the limitations of our GoF measure. We have calculated the coefficient of determination from the general form ($R^2 = 1 - \frac{SS_{residuals}}{SS_{total}}$). We acknowledge that this metric is limited if the correlation between the simulated and experimental data deviates from the identity line, so have calculated the RMSE as an additional metric. We have revised the figure to present both metrics, since the former tends to be intuitive despite its limitations. We hope that in conjunction with the graphical information which demonstrates the alignment of the data to the identity line, this gives the reader sufficient information to interpret the fit.

- Figure 3A: unit misses in the scale legend

We have added clarifying text in the captions specifying that this model is dimensionless.

Supplementary information:

Major:

- Line 31-32: ",the additional complexity does not change the qualitative aspects of the our conclusions". It is not clear how the statement is substantiated. Is this based on literature or did the authors actually test this.

We have added an example of the effect of adding additional complexity through the addition of basal non-growth-dependent lysis to the model in Supplemental Figure S4.

- Line 33-34 Did the authors try simulations with different initial values? And if so did that change the outcomes?

We have added examples of the effect of different initial values for the simulation in Figure S4. Initializing at different values does not change the relationship between the criterion and the direction of selection.

Minor:

First page.

- In the description of the model the description and preferably interpretation of the Hill coefficient (h and h_i) are missing.

We have added in a brief description of all parameters not already specified.

- Monod constant could be given as value or with a reference.

The Monod constant is reflective of the specific growth substrate and strain. In this case, we only use it to scale the nutrient concentration to produce a dimensionless model that does not specify one growth condition. More details on the model development can be found in Meredith et al. *Sci. Adv.* 2018, and we added a few descriptors to explain.

Throughout:

- use equation editor also in-line such that the symbols in the equations match the once in text. Currently hard to tell the difference between l ('el') and ι ('iota').

We have replaced all model variables and parameters in the text with the equation editor version.

Reviewer #2 (Remarks to the Author):

Comment:

The goal of the study is difficult to define and has not been clearly stated. However the study reveals certain interesting points:

- Variation in effective treatment doses and complex growth dynamics for mixed sensitive and resistant populations which is very interesting.
- Combination treatment impacts selection dynamics based on strain and drug-specific factors, including Bla production's private benefit, inhibitor suppression, and burden.
- Quantifying phenotypic bacterial responses for effective treatment strategies against resistant strains.

The author should define clear objective as well as importance of this study in terms of tackling resistance.

We thank the reviewer for recognizing the interesting aspects of our work and regret the confusion. The overall goal of the work is to understand which factors govern the selection dynamics in bacterial populations responding to combination β -lactam/Bla inhibitor treatment. Understanding these factors can improve treatment by differentiating which strains are good targets for such combinations and highlighting potential future targets for antimicrobial development. Specifically, our work makes the following actionable contributions:

1. We have shown that the privatization of the benefit of Bla production is critical in dictating the selection outcome of combination treatment, and that this property varies across known clinical isolates. Optimal combination doses can reduce resistance when treating strains with lower private benefit.
2. We have demonstrated how the private benefit can be estimated from growth curve measurements of clonal populations, which are much easier to perform than composition measurements. Taken with point 1) this estimation can be thought of as analogous to antibiotic susceptibility testing, where instead of predicting whether the population is *susceptible* to an antibiotic it predicts whether it is possible to *select against resistance* with a combination.
3. We have derived a general criterion for the selection outcome of combination treatment. This criterion highlights the importance of the burden and effective private benefit of resistance. These parameters, in addition to describing strains, present targets for further drug development. Our results suggest that increasing the burden of resistance (by developing agents that selectively target resistant cells) or increasing the suppression of the private benefit (by developing agents that can increase inhibitor penetration) would help combinations to select against resistance.

We have made revisions in the abstract, introduction, and discussion to better convey these points.

Reviewer #3 (Remarks to the Author):

This is a very interesting article that brings together experimental and modelling work on beta lactamase action, and in particular in the context of private benefit of periplasmic beta lactamase genes vs "public" benefit of cytoplasmic beta lactamase genes that become available to other cells following cell lysis. This is explored in the context of combined treatment with beta lactams and beta lactamase inhibitors.

While the paper is very interesting, I have serious reservations about the structure of the model and the underlying hypothesis of the study. The model considers a single population of resistant cells and single type of beta lactamase.

However, the paper has at its core the comparison between strains that produce differing amounts of cytoplasmic beta lactamase enzymes (which then enter the extracellular space following cell lysis) and periplasmic beta lactamase enzymes (which are only available to producer cells and presumably do not enter the extracellular space following cell lysis). The normal ecological argument that would be made in such a system is that there would be a balance between strains that produce a public good and those that consume it without producing it – i.e. 'cheater' strains that benefit from the cytoplasmic beta lactamase enzyme produced by other cells but which do not bear the cost of its production.

We thank the reviewer for recognizing the importance of understanding the ecological dynamics of population interactions that involve the public and private goods of Bla production. We should have made some aspects of our model, which considers resistant strains with different degrees of private benefit in the context of coculture with a sensitive population, clearer, and we have revised the text accordingly.

Of course this system is both more interesting and complicated because the public good is only available after the lysis of the producer cells, leading to potential also for kin selection dynamics. The model seems to describe none of these dynamics. At the very least I would expect to see two different beta lactamase genes in the model: the cytoplasmic and periplasmic genes each having their own equations and dynamics: periplasmic beta lactamase only benefitting the producer cells, while cytoplasmic beta lactamase benefitting the producer cells through internal concentration and all cells through external concentration.

This is pretty much essential for meaningful modelling of the system and appears to be missing from the model – in fact the model formulation does not appear to be a good description of either of these beta lactamase genes. The model then needs to be adapted so that it is clear how it could describe (i) cells that only produce cytoplasmic bla (ii) cells that only produce periplasmic bla (iii) cells that produce both (iv) ideally also (from an ecological point of view) cells that produce neither but which could benefit from cytoplasmic bla from other cells.

We regret the confusion, and we realize that we should have better explained our model formulation. Our model (Eqs S1–S10) accounts for the dynamics of a co-culture consisting of a sensitive strain and a Bla-producing strain. The sensitive strain does not produce a Bla but can benefit from Bla produced by the other strain.

The reviewer noted the different variants of Bla-producing strains. They are described by the parameter β_{min} in our model, which is the modulation (β) of the lysis rate of the resistant subpopulation in the absence of Bla inhibitor. If Bla provides complete protection for the producer, $\beta_{min} = 0$, and there is no antibiotic-dependent lysis of the producers. If Bla provides no protection, $\beta_{min} = 1$, and producers lyse at the same rate as the sensitive subpopulation. A periplasmic Bla would approach $\beta_{min} = 0$, while a cytoplasmic Bla would approach $\beta_{min} = 1$. However, in general, we expect the β_{min} values to fall on a spectrum, depending on the specific Bla and the host strain. In the presence of Bla inhibitor, β may be higher than β_{min} (Eq. S9).

As the reviewer suggests, strains can also produce multiple Bla enzymes at once, and this could be applicable to some of the clinical isolates we study in Figures 5 and 6. In this case, the Bla variable (b) can represent the combined effects of the different Bla variants, and β_{min} indicates the effective private benefit conferred by all Bla enzymes.

As currently formulated, the model in the paper doesn't appear to be able to describe any of the cells in the paper. And as soon as it is thought through on these terms, it becomes clear that single population dynamics are insufficient to decouple private vs public goods advantages of the system.

For example, cells that only produce periplasmic bla would benefit from their neighbours who produce cytoplasmic bla but not vice versa. One of the advantages of the model is that it could be used to examine mixed populations. The biggest weakness of the paper is that it is trying to understand a complex ecological idea from single strains – and this impacts both on the experimental and modelling work.

Again, we regret the confusion. All our analysis on evolutionary dynamics was done in co-cultures. In the original submission, each co-culture system consists of a sensitive strain and a Bla-producing strain (engineered or natural). In the revised submission, we also included additional experiments that show the robustness of our conclusion when we include additional sensitive strains in the co-culture (see Response to reviewer #1).

It is possible the confusion might have resulted from the fact that we estimated model parameters for the clinical isolates by using a single-population model (modifications to base model detailed in Eqs S16-S18) and clonal cultures (Figure 5A–D). Our results, however, show that we can indeed use clonal population dynamics to determine parameters critical for coculture dynamics. In Figures 5E and 6, we present β_{min} values, estimated from single population dynamics, for a set of clinical isolates, and show how they coarsely predict the result of coculturing these isolates with a sensitive strain (or a community of sensitive strains). Our results show that although the system we are ultimately trying to understand is complex, this key outcome can be analyzed and reproduced in simpler contexts.

We hope that this clarifies our model structure and the scope of our work, and have introduced text in the manuscript to make it clearer to readers.

Other specific comments:

Abstract:

We note that both the title and abstract highlights the importance of the “private” benefit of bla production. Implicit in this statement is that there may be a contrasting public “benefit” argument (or even public goods game problem) that frames this private benefit. The Abstract needs the public context in order for that private argument to be made. Please can a relevant sentence be added in order to ensure correct logical flow.

We have inserted language specifying that Bla production provides private benefit to the resistant cells and public benefit that sensitive cells can exploit.

Results

L85-125

This section is somewhat confusing because it mixes material that belongs in the Introduction (including reference to previous work, and material on private vs public benefits of Bla proteins) with results produced in this paper. This needs to be remedied. Relevant introductory material needs to be moved to the Introduction and the Results should ONLY describe the experimental results from this study. The Introduction would also massively benefit e.g. by discussion of periplasmic vs cytoplasmic Bla enzymes, as well as some further description of the likely consequences of private vs public goods in this context (to benefit those readers who might have a more molecular rather than ecological understanding of drug resistance).

We thank the reviewer for their helpful suggestions. We have revised the beginning of the Results and the Introduction to better delineate the sections and include more discussion of public and private goods.

L98 statement is vague. Can you specify in what way the responses are “profound”? It is actually difficult to understand Figure 1a on the basis of what is written both in the Results text and the figure caption so some more explicit explanation for the reader is needed.

We regret the confusion. We have revised the text in this section to clarify the purpose of this figure in demonstrating differences in responses to combination doses depending on the strain or Bla inhibitor used. The differences we highlight here include the necessary concentrations to suppress the population and the degree of synergy and antagonism between the beta-lactam and Bla inhibitor.

L112-114. The reader does not know which strains have more private benefit: there is insufficient description of the strains in the Results to be able to clearly follow the argument. I appreciate that it is sometimes harder to write papers when Methods are at the end (rather than before Results) but some brief synopsis of what strains are involved is needed in the Results section to make the Results interpretable.

We regret the confusion. We have added more notations to specify the strains with more private benefit. We hope that in conjunction with other edits made in this section that the flow of the logic is now clearer for the reader.

Reviewers' comments:

Reviewer #1 (Remarks to the Author):

I would like to thank the authors for the additional work done and addressing all my remarks in the previous review.

My only suggestion would be to write out the derivation of the new criterion with HGT so that it can be better understood. Also please note down that n_0 r_0 etc. are the values at $t=0$. One would assume this, but it is not explicitly mentioned.

Reviewer #1 (Remarks on code availability):

Code is available.

Reviewer #4 (Remarks to the Author):

I was not one of the original reviewers; I was asked to evaluate whether the authors have addressed the comments raised by Reviewer #3. In my view, the original comments have been generally well addressed. However, I do have some additional concerns. I realise this is annoying - I have kept these to a minimum and only bring them up because I think they are critical to the central message of the paper.

Reviewer #3's comments:

Reviewer #3's main criticism of the paper is the model structure. The model in the manuscript is formulated in terms of sensitive and resistant populations. In the reviewer's view, the model should have included multiple resistant populations differing in the extent to which resistance is a private vs public good.

My view is that reviewer #3's interpreted the central question of the paper differently from what the authors intended. The reviewer's comments suggest that they see the modelling as attempting to understand the competition between different resistance strategies (public vs private). However, my interpretation is that the purpose of the model was to explore how antibiotics + beta-lactamase inhibitors affect selection for resistance, depending on the extent to which the benefit of resistance is private vs public. I think this is a different question and adequately addressed by the model structure in the manuscript.

I think the changes to the manuscript have probably clarified this issue sufficiently, as the purpose of the modelling was clear in my reading of the paper. That said, I do think some of the language is potentially confusing on this point, particularly the use of the term "privatization," including in the title - because this implies a switch from a public good strategy to a private benefit strategy, which is not, as reviewer #3 points out, addressed by the model. Another example is in the discussion (L394): "this insight is reminiscent of studies on other systems that demonstrate how privatization of a public good can ensure

the maintenance of a cooperative trait.” This is potentially a little misleading, particularly as the cited studies do look at the type of models suggested by reviewer #3. I think clarifying what the authors mean by “privatization” would be helpful.

The specific comments of the reviewer are well addressed.

My comments:

Overall, I think the paper addressed an interesting question, is clearly written and a very nice example of combining modelling and experiments. I think some of the results are very convincing, but have concerns about the presentation/interpretation of some of the modelling results.

1. One of the key results of the paper is that the selection outcome depends on the private benefit of Bla (as opposed to the public benefit) - e.g. L391: “the function of Bla is often considered a public good, due to its ability to degrade beta-lactams. Our results reveal the critical importance of the private benefit of Bla in dictating the outcome of selection dynamics in response to combination treatment (Eq. 1, Figure 3).”

The authors derive the criterion for resistance selection assuming a fixed antibiotic concentration. In the model, the public benefit of resistance works through lowering the antibiotic concentration (either through absorption by live resistant cells or through free Bla from lysed resistant cells). Thus, under the assumptions of fixed antibiotic concentration, there is no possible public benefit and the model simplifies to a model of private resistance. This is not a problem in itself, but this should be made very clear in the text. The result that selection for resistance in this model depends on the balance of cost and benefit is expected and well known.

The authors then check how well this criterion performs against full model simulations with randomly picked parameter values (Figure 3C) and conclude the criterion is an excellent predictor of simulation results. But whether the criterion is a good predictor or not depends entirely on the parameter ranges the authors sample from, and without a sense of what the plausible parameter range is, it is very difficult to interpret this result. For example, if the authors sampled from a range including a higher public benefit (i.e. higher κ_b or ϕ), the criterion would presumably increasingly over-estimate the success of resistance.

I appreciate it is very difficult to know what parameter ranges are reasonable. I think under such conditions, a helpful strategy is to explicitly explore the conditions under which the simple criterion stops being a good predictor of the outcome of the more complex model. For example, in this case, exploring the effect of increasing the public benefit of resistance on how well the criterion predicts outcomes of the model. Alternatively/additionally, the authors could use their parameter estimation to inform a plausible parameter range for exploration.

Fundamentally, my concern is that one of the central messages from the modelling is that selection for resistance depends on the private as opposed to public benefit of resistance (without this opposition to public benefit, the result that selection depends on the benefit of resistance is already well understood).

The contrast of public vs private benefit is not sufficiently explored in the model to allow this conclusion.

2. If at all possible, I would include the model equations in the main text. The modelling results are not really interpretable without the equations, so it's quite a lot of the paper that cannot be understood without the supplement.

Point-by-point responses to reviewers' comments

REVIEWER COMMENTS

Reviewer #1 (Remarks to the Author):

I would like to thank the authors for the additional work done and addressing all my remarks in the previous review.

My only suggestion would be to write out the derivation of the new criterion with HGT so that it can be better understood. Also please note down that n_0 n_r etc. are the values at $t=0$. One would assume this, but it is not explicitly mentioned.

We thank the reviewer for their efforts to review the revised paper. We have expanded on the derivation of the new criterion in the supplement and added the requested definition of the initial parameter values in both the supplement and the main text.

Reviewer #4 (Remarks to the Author):

I was not one of the original reviewers; I was asked to evaluate whether the authors have addressed the comments raised by Reviewer #3. In my view, the original comments have been generally well addressed. However, I do have some additional concerns. I realise this is annoying - I have kept these to a minimum and only bring them up because I think they are critical to the central message of the paper.

We thank the reviewer for providing constructive feedback on the work.

Reviewer #3's comments:

Reviewer #3's main criticism of the paper is the model structure. The model in the manuscript is formulated in terms of sensitive and resistant populations. In the reviewer's view, the model should have included multiple resistant populations differing in the extent to which resistance is a private vs public good.

My view is that reviewer #3's interpreted the central question of the paper differently from what the authors intended. The reviewer's comments suggest that they see the modelling as attempting to understand the competition between different resistance strategies (public vs private). However, my interpretation is that the purpose of the model was to explore how antibiotics + beta-lactamase inhibitors affect selection for resistance, depending on the extent to which the benefit of resistance is private vs public. I think this is a different question and adequately addressed by the model structure in the manuscript.

I think the changes to the manuscript have probably clarified this issue sufficiently, as the purpose of the modelling was clear in my reading of the paper. That said, I do think some of the language is potentially confusing on this point, particularly the use of the term "privatization," including in the title - because this implies a switch from a public good strategy to a private benefit strategy, which is not, as reviewer #3 points out, addressed by the model. Another example is in the discussion (L394): "this insight is reminiscent of studies on other systems that demonstrate how privatization of a public good can ensure the maintenance of a cooperative trait." This is potentially a little misleading, particularly as the cited studies do look at the type of models suggested by reviewer #3. I think clarifying what the authors mean by "privatization" would be helpful.

The specific comments of the reviewer are well addressed.

We thank the reviewer for recognizing our efforts to clarify the model structure in response to Reviewer #3's comments and for appreciating the central question we attempted to address with our modeling.

We also appreciate the reviewer's comment on the term "privatization", which indeed has connotations that we do not intend to convey. We have revised the manuscript to reference the private component of the mixed good of Bla production rather than the verb usage of privatization where appropriate, including in the title.

My comments:

Overall, I think the paper addressed an interesting question, is clearly written and a very nice example of combining modelling and experiments. I think some of the results are very convincing, but have concerns about the presentation/interpretation of some of the modelling results.

We thank the reviewer for recognizing the interest of the question and results. We address their concerns, with corresponding revisions, below.

1. One of the key results of the paper is that the selection outcome depends on the private benefit of Bla (as opposed to the public benefit) - e.g. L391: "the function of Bla is often considered a public good, due to its ability to degrade beta-lactams. Our results reveal the critical importance of the private benefit of Bla in dictating the outcome of selection dynamics in response to combination treatment (Eq. 1, Figure 3)."

The authors derive the criterion for resistance selection assuming a fixed antibiotic concentration. In the model, the public benefit of resistance works through lowering the antibiotic concentration (either through absorption by live resistant cells or through free Bla from lysed resistant cells). Thus, under the assumptions of fixed antibiotic concentration, there is no possible public benefit and the model simplifies to a model of private resistance. This is not a problem in itself, but this should be made very clear in the text. The result that selection for resistance in this model depends on the balance of cost and benefit is expected and well known.

We regret the confusion. We have revised our manuscript to clarify the rationale of the simplified form of the criterion.

The more general form of the criterion presented before simplifying assumptions (Equation S13, $1 - \beta > \frac{1-\alpha}{l/g}$), does account for these dynamics. In this form, which is now presented as Eq 1, upon antibiotic degradation, public benefit reduces l , the effective lysis rate.

The full model likewise considers the public benefit, and as discussed in the results, the lack of complete alignment between the criterion and the selection dynamics is likely due to the further simplifications we made in deducing the criterion for the limiting case (where both the antibiotic and the inhibitor are in excess, e.g. at time 0). However, this criterion for the limiting case has the advantage of being uniquely defined by parameters that can be quantified using clonal populations.

We agree with the reviewer that selection depends on the cost-benefit balance in general. However, the details of the cost-benefit balance are system-specific. Our study established criteria to predict the selection dynamics resulting from combinations of beta-lactams and Bla inhibitors. Despite the wide use of such combination treatment, the criteria we derived are new, and they resolve apparently conflicting results described in the literature.

When considering Bla-mediated evolutionary dynamics, past analyses have focused on two aspects: the burden of resistance, as well as its public benefit. Our criteria reveal the critical role of the private benefit, which we also show to be more variable among clinical isolates than the burden of resistance (as reflected by much wider distribution of β_{min} than μ).

The authors then check how well this criterion performs against full model simulations with randomly picked parameter values (Figure 3C) and conclude the criterion is an excellent predictor of simulation results. But whether the criterion is a good predictor or not depends entirely on the parameter ranges the authors sample from, and without a sense of what the plausible parameter range is, it is very difficult to interpret this result. For example, if the authors sampled from a range including a higher public benefit (i.e. higher kappa_b or phi), the criterion would presumably increasingly over-estimate the success of resistance.

I appreciate it is very difficult to know what parameter ranges are reasonable. I think under such conditions, a helpful strategy is to explicitly explore the conditions under which the simple criterion stops being a good predictor of the outcome of the more complex model. For example, in this case, exploring the effect of increasing the public benefit of resistance on how well the criterion predicts outcomes of the model. Alternatively/additionally, the authors could use their parameter estimation to inform a plausible parameter range for exploration.

We thank the reviewer for this insightful suggestion for tackling simulations given parameters without clearly knowable ranges. We have repeated the analysis in Figure 3C in the new Figure S5C using the ranges for φ_{max} and κ_b from the parameter estimation from the clinical isolates described in Figure S7. These ranges are sufficiently wide to describe the dynamics seen in experimental data: the estimated parameters do not cluster at the upper end of the range boundary (Figure S7). Notably, as is consistent with the results from sensitivity analysis from Figure S3 for φ_{max} and κ_b , increasing the ranges for these parameters does not affect the predictive ability of the criterion.

We recognize that this can be unintuitive, as at first glance this may suggest that the public benefit is irrelevant to the selection dynamics. An important point regarding the criterion is that the data in Figure 3C collapses the complex dose-response landscapes seen in Figure 2D and 3A into a single value, which describes the minimum possible resistant fraction for that landscape. Higher public benefit does impact the selection dynamics in this system: it reduces the number of doses at which resistant cells are selected for, as the period during which resistant cells may dominate will be shorter. It also increases cell survival: in Figure S5C, when a higher bound is used for the public benefit parameters, there are a greater number of empty circles, where no treatment concentrations achieved suppression. We have added dose-response landscapes to demonstrate this concept in Figure S5A and S5B (also see below).

However, whether it's possible to select against the resistant population using combination treatment, at a dose that suppresses the population, depends primarily on the magnitude of the private benefit and its suppression by the inhibitor. This is what the criterion tests for, and what

our experimental data supports. Because both subpopulations experience the public benefit, increasing it does not affect the maximum differential advantage of the resistant strain. This is why, in the sensitivity analysis in Figure S3, we find that the resistant fraction at a suppressive dose is insensitive to the value of the parameters that govern the public benefit.

In addition to the new analysis, we have added text referring to these findings in the Results and in the figure caption. We hope this clarifies the different roles that the public and private benefits conferred by Bla production play in the population dynamics in this system.

Revised Supplemental Figure S5. Increasing public benefit parameter ranges affects simulated cell survival and selection dynamics, but does not change the minimum achievable resistant fraction for a given landscape.

- A. Increasing extracellular Bla-mediated antibiotic degradation increases the range of concentrations that the population can survive (top, heatmap color indicates final cell density for each dose) and at which sensitive cells are selected for (bottom, heatmap color indicates final resistant fraction for

each dose). However, it does not affect the magnitude of the minimum achievable resistant fraction (bottom).

- B. Increasing intracellular Bla-mediated antibiotic degradation increases the range of concentrations that the population can survive (top, color indicates cell density) and at which sensitive cells are selected for (bottom, color indicates resistant fraction). However, it does not affect the magnitude of the minimum achievable resistant fraction (bottom).
- C. Increasing the range of intra- and extracellular Bla-mediated antibiotic degradation parameters when simulating 10,000 strains with random parameter sets does not affect the ability of the criterion to predict the minimum achievable resistant fraction for each strain (dot color). However, an increased range in public benefit parameters does increase the number of simulated strains where no dose was high enough to suppress the population (empty circles).

Fundamentally, my concern is that one of the central messages from the modelling is that selection for resistance depends on the private as opposed to public benefit of resistance (without this opposition to public benefit, the result that selection depends on the benefit of resistance is already well understood). The contrast of public vs private benefit is not sufficiently explored in the model to allow this conclusion.

We appreciate the reviewer raising this concern about the modeling, which has guided us to provide further analyses and clarifications, as well as corresponding revisions in the manuscript, as detailed above. We hope these clarifications and analyses will address the reviewer's concern.

2. If at all possible, I would include the model equations in the main text. The modelling results are not really interpretable without the equations, so it's quite a lot of the paper that cannot be understood without the supplement.

We thank the reviewer for the feedback. They raise a good point and we have added the base model equations to the Methods.

REVIEWERS' COMMENTS

Reviewer #4 (Remarks to the Author):

I thank the authors for their careful consideration and response to my comments. I think the additions have strengthened and clarified the paper, and that this work will be an interesting and valuable addition to the literature and requires no further revision.